# CAST: Cross-Attention in Space and Time
# for Video Action Recognition

**Dongho Lee**[*]         **Jongseo Lee**[*]         **Jinwoo Choi**[†]

**Kyung Hee University, Republic of Korea**
{kide004, jong980812, jinwoochoi}@khu.ac.kr

## Abstract

Recognizing human actions in videos requires spatial and temporal understanding. Most existing action recognition models lack a balanced spatio-temporal understanding of videos. In this work, we propose a novel two-stream architecture, called Cross-Attention in Space and Time (CAST), that achieves a balanced spatio-temporal understanding of videos using only RGB input. Our proposed bottleneck cross-attention mechanism enables the spatial and temporal expert models to exchange information and make synergistic predictions, leading to improved performance. We validate the proposed method with extensive experiments on public benchmarks with different characteristics: EPIC-KITCHENS-100, Something-Something-V2, and Kinetics-400. Our method consistently shows favorable performance across these datasets, while the performance of existing methods fluctuates depending on the dataset characteristics. The code is available at https://github.com/KHU-VLL/CAST.

## 1 Introduction

To accurately recognize human actions in videos, a model must understand both the spatial and temporal contexts. A model that lacks fine-grained spatial understanding is likely to fail in predicting the correct action. For example, as shown in Figure 1 (a), a model that understands temporal context such as hand motion across frames but not the fine-grained spatial context may confuse whether an object in the hand a *ketchup*, or a *cheese*, or a *milk carton*. Consequently, the model fails to predict the correct action, *Put down a cheese*. Similarly, a model that lacks temporal context understanding may also fail to predict the correct action. In Figure 1 (b), let us suppose a model understands spatial context but does not understand temporal context, e.g., the model is confused about whether the hand is moving from outside the fridge to the inside or vice versa. Then the model fails to predict the correct action of *Take out a sauce*. Therefore, for accurate action recognition, models need to comprehend both the spatial and temporal contexts of videos.

Despite the recent progress in action recognition through the use of Transformers [60, 11, 3], achieving a balanced spatio-temporal understanding remains a challenging problem. Compared to images, the additional temporal dimension in videos makes spatio-temporal representation learning computationally intensive and requires a significant amount of training data [3]. Consequently, most action recognition models lack a balanced spatio-temporal understanding of videos. Notably, models that perform well on static-biased [32, 8, 50] datasets, such as Kinetics-400, may not perform as well on temporal-biased [3, 28] datasets, such as Something-Something-V2, and vice versa. For instance, as shown in Figure 4 (a), on the EPIC-KITCHENS-100 dataset, VideoMAE [56] outperforms ST-Adapter [42] on the verb prediction task, while ST-Adapter outperforms VideoMAE on the noun prediction task. Similarly, BEVT [62] outperforms AIM [72] on the Something-Something-V2

---

[*]Equally contributed first authors.

[†]Corresponding author.

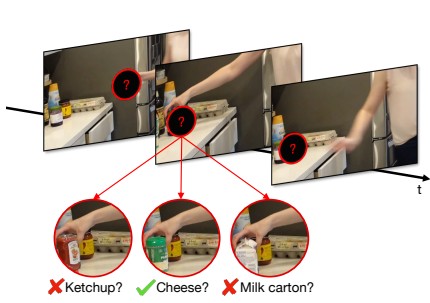
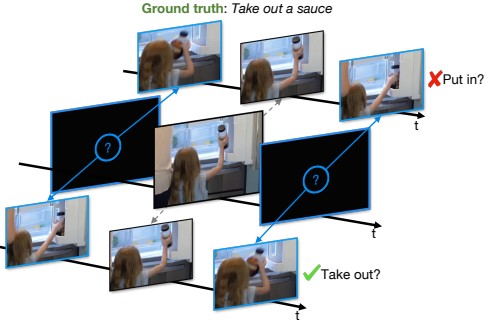

(a) Importance of spatial understanding

(b) Importance of temporal understanding

Figure 1: **The importance of spatio-temporal understanding.** If a model lacks fine-grained spatial understanding, the model may predict an incorrect action. E.g., the model fails to predict *Put down a cheese* in (a) due to subtle appearance differences between the objects. On the other hand, if a model lacks temporal context understanding, the model may predict an incorrect action. E.g., the model fails to predict *Take out a sauce* in (b) due to the ambiguity of the action. Therefore, both spatial and temporal understanding are crucial in action recognition. Best viewed with zoom and color.

dataset, while BEVT underperforms AIM on the Kinetics-400 dataset. We observe a similar trend for other methods as reported in Table 2.

One possible solution to the challenge of balanced spatio-temporal understanding is to use multi-modal learning. For example, two-stream networks [51, 14] employ both RGB and optical flow streams to learn both spatial and temporal contexts. However, this approach can be computationally expensive due to optical flow estimation.

In this work, we introduce a two-stream architecture, Cross-Attention in Space and Time (CAST), to address the challenge of balanced spatio-temporal understanding using only RGB input. In Figure 2, we show a high-level illustration of the proposed method. Our architecture employs two expert models - a spatial expert model and a temporal expert model - which exchange information to make a synergistic collective prediction. We realize the information exchange by cross-attention between the two experts. We empirically validate that placing cross-attention in a bottleneck architecture facilitates more effective learning. To validate the effectiveness of the proposed method, we conduct extensive experiments on multiple datasets with distinct characteristics, including the temporal-biased Something-Something-V2, static-biased Kinetics-400, and fine-grained EPIC-KITCHENS-100. Our results demonstrate that CAST achieves balanced spatio-temporal understanding and shows favorable performance across these different datasets.

In this work, we make the following significant contributions.

- We introduce a two-stream architecture, CAST, which addresses the challenge of *balanced spatio-temporal understanding* that has been largely overlooked by previous works.
- We conduct extensive experiments on multiple datasets with distinct characteristics to demonstrate the effectiveness of CAST. In terms of *balanced* spatio-temporal understanding, CAST shows favorable performance, while existing methods show more imbalanced performance.
- We conduct an extensive ablation study and analysis to validate the design choices of the proposed method. We show that employing *spatial expert* and *temporal expert* and placing *cross-attention in a bottleneck* architecture is crucial for achieving effective spatio-temporal representation learning.

## 2   Related Work

**Video Action Recognition.**   CNN-based approaches have been widely used for action recognition, including 2D CNNs [61, 74, 33, 52, 27], 3D CNNs [58, 5, 59, 64, 13], 2D and 1D separable CNNs [59, 70], or two-stream CNNs [14, 15]. These methods have achieved great progress thanks to the strong inductive biases. Recently, Transformer-based approaches [1, 3, 21, 43, 68, 12, 71] become popular in the community due to the long-term context modeling capabilities. Similar to the two-stream CNNs, we propose a two-stream transformer architecture consisting of two expert models: a spatial expert and a temporal expert. However, unlike traditional two-stream CNNs, we use RGB input only, instead of RGB and flow.

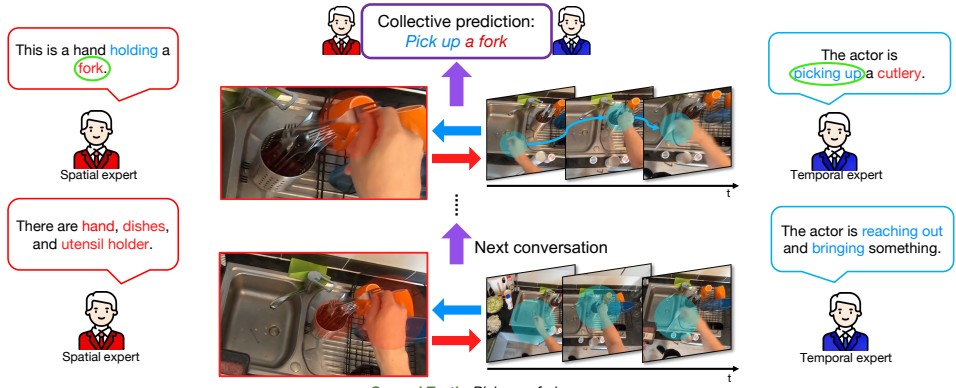

Figure 2: **High-level illustration of the proposed method.** In this work, we employ spatial and temporal expert models. The two experts exchange information with each other using cross-attention. Initially, the experts may predict incorrect actions due to the lack of information. For example, the temporal expert may predict *reach out to something* while the ground truth is *Pick up a fork*. Similarly, the spatial expert may predict *utensil holder* instead of *fork* in the shallower layers. However, after using cross-attention to exchange information multiple times, the proposed method can collectively predict the correct action *Pick up a fork*. Best viewed with zoom and color.

**Cross-attention.**  Cross-attention has been widely utilized in multi-modal learning to facilitate information exchange between different modalities such as audio, visual, and text [34, 67, 40, 30, 18]. Recently, cross-attention between different views of the same video has shown impressive results [71, 75, 6, 26]. Similar to these, we propose a cross-attention method using a single RGB input, but with two distinct expert models: a spatial expert and a temporal expert. The two experts attend to each other through cross-attention to achieve a balanced spatio-temporal understanding.

**Foundation model.**  Trained on web-scale datasets using self-supervised learning, foundation models [25, 4, 48, 45, 44] are highly adaptable and versatile. Foundation models show impressive performance on various tasks in computer vision [65, 62], natural language processing [49, 57], and audio recognition [17]. In this work, we employ CLIP [44] as our spatial expert as it shows impressive performance on more than 30 computer vision tasks.

**Parameter-efficient transfer learning.**  Although the "pre-training and fine-tuning" paradigm with strong foundation models has demonstrated impressive performance on several computer vision tasks, it is computationally expensive and often unnecessary to fine-tune the full model [72]. Several works have demonstrated that learning only a small subset of parameters and keeping the remaining parameters frozen is effective for NLP tasks [23, 29] and computer vision tasks [36, 66, 54, 46, 47]. Extending image foundation models by adding adapter architectures has shown favorable performance on action recognition [35, 72, 42]. The proposed method also employs adapter architecture with cross-attention between two experts. We empirically demonstrate that the proposed method outperforms existing adapter-based video models in terms of achieving balanced spatio-temporal understanding.

## 3  Method: Cross-Attention in Space and Time

We introduce CAST, a method for balanced spatio-temporal representation learning for action recognition, as shown in Figure 3. We employ frozen spatial and temporal expert models that can be any vision transformer, consisting of 12 transformer blocks each. To facilitate information exchange between the experts, we introduce the bottleneck cross-attention in space and time (B-CAST) module on top of the frozen layers. This module enables the experts to exchange information and learn more balanced spatio-temporal contexts than separate experts. To improve adaptation to downstream tasks, we use adapter layers with a small number of learnable parameters, following AIM [72]. In the following subsections, we provide a detailed description of each component of our proposed CAST.

### 3.1  Input embeddings

CAST takes only RGB videos as inputs. The input is a mini-batch of videos, $\mathbf{I} \in \mathbb{R}^{B \times 2T \times H \times W \times C}$, consisting of $B$ videos of $2T$ frames, $H \times W$ spatial dimensions, and $C$ channels. We apply patch tokenization to the input videos for the spatial expert and the temporal expert. For the spatial expert, we decompose every even frame of each video in $\mathbf{I}$ into $N$ non-overlapping patches of $p \times p$ pixels [11].

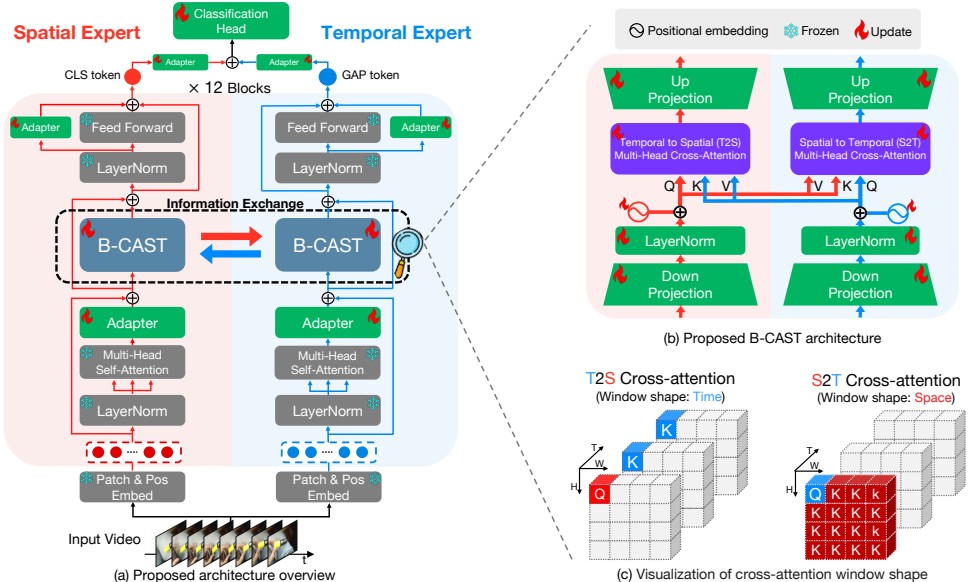

(a) Proposed architecture overview

(b) Proposed B-CAST architecture

(c) Visualization of cross-attention window shape

Figure 3: **Overview of CAST.** (a) CAST employs frozen spatial and temporal expert models. On top of the experts, we add a cross-attention module B-CAST to enable the exchange of information between the two experts. Additionally, we employ adapters with a small number of learnable parameters to the experts for better adaptation. (b) The proposed B-CAST consists of temporal-to-spatial (T2S) and spatial-to-temporal (S2T) cross-attentions to allow for a better understanding of the spatio-temporal features in the video data. For efficient and effective learning, we incorporate cross-attention into the bottleneck adpater. We employ separate position embedding for each expert. (c) We visualize T2S and S2T cross-attentions. Given a query, the model attends along the temporal axis only in T2S while the model attends along the spatial axes only in S2T.

Then we pass the patches through a frozen linear layer and add position embeddings to obtain spatial embeddings, $\mathbf{X}_s \in \mathbb{R}^{BT \times N \times D}$. For the temporal expert, we decompose every two frames of each video in $\mathbf{I}$ into $2 \times p \times p$ pixels non-overlapping tubes [1]. Then we pass the tubes through a frozen linear layer and add position embeddings to obtain temporal embeddings, $\mathbf{X}_t \in \mathbb{R}^{B \times TN \times D}$.

## 3.2 CAST architecture

The model architecture of each expert is the same as the ViT except for adapters and the B-CAST module. All the other parameters are frozen, while the adapter and B-CAST parameters are learnable.

For completeness, we first define the operations used and then describe the entire model architecture. Given an input $\mathbf{X}$, we define Multi-Head Self Attention (MHSA) operation as follows:

$$\text{MHSA}(\mathbf{X}) = \text{Softmax}((\mathbf{X}\mathbf{W}_Q)(\mathbf{X}\mathbf{W}_K)^\top)(\mathbf{X}\mathbf{W}_V), \tag{1}$$

$\mathbf{W}_Q$, $\mathbf{W}_K$, and $\mathbf{W}_V$ are the query, key, and value projection matrices, respectively. We also define the adapter operation with linear down and up projection matrices $\mathbf{W}_D$ and $\mathbf{W}_U$ as follows:

$$\text{ADAP}(\mathbf{X}) = \sigma(\mathbf{X}\mathbf{W}_D)\mathbf{W}_U, \tag{2}$$

where $\sigma(\cdot)$ is the GELU activation function [20].

For each attention block $l$, we apply independent Multi-Head Self Attention (MHSA) for each expert along with a skip connection as follows:

$$\mathbf{Y}^{(l)} = \mathbf{X}^{(l)} + \text{ADAP}(\text{MHSA}(\text{LN}(\mathbf{X}^{(l)}))) + \text{MHSA}(\text{LN}(\mathbf{X}^{(l)})), \tag{3}$$

where $\text{LN}(\cdot)$ denotes the Layer Normalization operation. The spatial path undergoes spatial attention, while the temporal path undergoes space-time attention following TimeSformer [3].

As shown in Figure 3 (b), to exchange information between the two experts, we apply the B-CAST operation $\Phi(\cdot)$ to $\mathbf{Y}_{e_1}$ and $\mathbf{Y}_{e_2}$ from the expert $e_1$ and $e_2$ as follows along with a skip connection:

$$\mathbf{B}^{(l)} = \mathbf{Y}^{(l)} + \Phi(\mathbf{Y}_{e_1}^{(l)}, \mathbf{Y}_{e_2}^{(l)}). \tag{4}$$

We describe the B-CAST operation $\Phi(\cdot)$ in detail in Section 3.3.

Finally, we pass the output, denoted as $\mathbf{B}^{(l)}$, through a two-layer feed forward network (FFN) [11] with the GELU activation function in between the layers and another adapter to obtain the next layer input $\mathbf{X}^{(l+1)}$ as follows along with a skip connection:

$$\mathbf{X}^{(l+1)} = \mathbf{B}^{(l)} + \mathrm{FFN}(\mathrm{LN}(\mathbf{B}^{(l)})) + \mathrm{ADAP}(\mathrm{LN}(\mathbf{B}^{(l)})). \tag{5}$$

**Classification head.** To produce the final prediction, we need to aggregate the outputs of both spatial and temporal experts. For the spatial expert, we average the frame-level class tokens from the last attention block, $\mathbf{X}_s^{(12)}$, to obtain a single class token. We denote this operation as $\mathrm{CLS}(\cdot)$. To obtain temporal expert features, we aggregate all the tokens from the last attention block of the temporal expert, $\mathbf{X}_t^{(12)}$, using the global average pooling $\mathrm{GAP}(\cdot)$ operation. Then we add the adapter output of the CLS token and the adapter output of the GAP token to produce a fused token $\mathbf{Z}$:

$$\mathbf{Z} = \mathrm{ADAP}(\mathrm{CLS}(\mathbf{X}_s^{(12)})) + \mathrm{ADAP}(\mathrm{GAP}(\mathbf{X}_t^{(12)})). \tag{6}$$

Finally, we feed the fused token $\mathbf{Z}$ a classification layer followed by a softmax function to obtain the predicted class probabilities. We train the model using the standard cross-entropy loss.

### 3.3 B-CAST module architecture

**Multi-Head Cross-Attention.** Multi-Head Cross-Attention (MHCA) is a variant of the MHSA operation (1), where query tokens come from one expert ($e_1$) and key and value tokens come from another expert ($e_2$). This allows the experts to exchange information and benefit from the strengths of each other. We define the MHCA operation as follows:

$$\mathrm{MHCA}(\mathbf{Y}_{e_1}, \mathbf{Y}_{e_2}) = \mathrm{Softmax}((\mathbf{Y}_{e_1}\mathbf{W}_Q)(\mathbf{Y}_{e_2}\mathbf{W}_K)^\top)(\mathbf{Y}_{e_2}\mathbf{W}_V), \tag{7}$$

where $\mathbf{W}_Q$, $\mathbf{W}_K$, and $\mathbf{W}_V$ are learnable query, key, and value parameter matrices respectively.

**Temporal-to-Spatial Cross-Attention.** In Temporal-to-Spatial (T2S) cross-attention, query tokens come from the spatial expert $s$, and key and value tokens come from the temporal expert $t$: $\mathrm{MHCA}(\mathbf{Y}_s^{(l)}, \mathbf{Y}_t^{(l)})$. We depict the attention window in Figure 3 (c). Given a query, the model attends along the temporal dimension only. By using T2S cross-attention, the spatial expert can learn to attend to temporal features from the temporal expert. T2S MHCA leads to capturing spatio-temporal dependencies and improves the model performance in action recognition.

**Spatial-to-Temporal Cross-Attention.** In Spatial-to-Temporal (S2T) cross-attention, query tokens come from the temporal expert $t$, and key and value tokens come from the spatial expert $s$: $\mathrm{MHCA}(\mathbf{Y}_t^{(l)}, \mathbf{Y}_s^{(l)})$. We illustrate the attention window in Figure 3 (c). Given a query, the model attends along the spatial dimension only. By using S2T cross-attention, the temporal expert can attend to fine-grained spatial features from the spatial expert. S2T MHCA leads to a more balanced spatio-temporal understanding and improves the performance in fine-grained action recognition.

**Bottleneck Cross-Attention in Space and Time.** To achieve efficient and effective learning, we incorporate the T2S and S2T MHCA into bottleneck-shaped adapters. We illustrate B-CAST architecture in Figure 3 (b). We plug the MHCA modules into adapters and add new learnable positional embeddings for each MHCA. We define the B-CAST operation for T2S $\Phi_S(\cdot)$ as follows:

$$\Phi_S(\mathbf{Y}_s^{(l)}, \mathbf{Y}_t^{(l)}) = \sigma(\mathrm{MHCA}(\mathbf{E}_s + \mathrm{LN}(\mathbf{Y}_s^{(l)}\mathbf{W}_{D,s}), \mathbf{E}_t + \mathrm{LN}(\mathbf{Y}_t^{(l)}\mathbf{W}_{D,t})))\mathbf{W}_{U,s}, \tag{8}$$

where $\mathbf{W}_{D,s}$ and $\mathbf{W}_{U,s}$ are linear down- and up-projection matrices for the spatial expert, and $\mathbf{E}_s$ and $\mathbf{E}_t$ are new positional embeddings for the spatial and temporal experts, $\sigma(\cdot)$ is the GELU activation function, respectively. We can define the B-CAST operation for S2T, $\Phi_T(\cdot)$ in a similar manner. The output of B-CAST goes into a feed forward network using (5). Our empirical validation shows that the B-CAST architecture is efficient and effective. (See Table 1.)

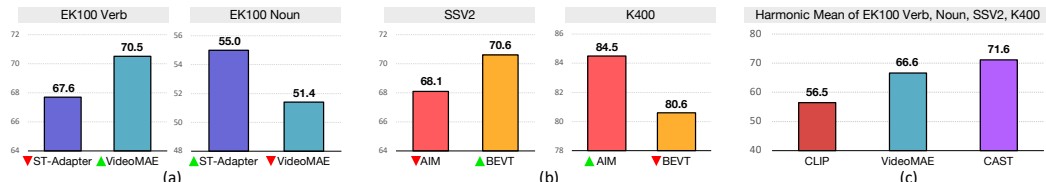

Figure 4: **Balanced spatio-temporal understanding performance.** We visualize the action recognition accuracies of existing methods and the proposed method. (a) We show the Top-1 accuracies of ST-Adapter and VideoMAE on the EK100 verb and noun prediction tasks. (b) We show the Top-1 accuracies of AIM and BEVT on the SSV2, and K400. (c) For each method, we show the harmonic mean of Top-1 accuracies on the EK100 noun, EK100 verb, SSV2, and K400. CAST shows a more balanced spatio-temporal understanding capability compared to the existing methods. Best viewed with zoom and color.

## 4 Experimental Results

In this section, we present the experimental results that answer the following research questions: (1) Do existing methods show a balanced spatio-temporal understanding of videos? (Section 4.3) (2) What are the ingredients for a balanced spatio-temporal understanding? (Section 4.3) (3) Is the proposed method effective? (Section 4.3, Section 4.4) (4) How can we effectively combine spatial and temporal models to achieve such balance? (Section 4.5) (5) Does the proposed method outperform state-of-the-art methods in terms of balanced spatio-temporal understanding? (Section 4.6) To this end, we first provide details about the datasets and implementation in Section 4.1 and Section 4.2, respectively.

### 4.1 Datasets

**Action recognition.** We evaluate the CAST on two public datasets for conventional action recognition: Something-Something-V2 (SSV2) [19] and Kinetics-400 (K400) [24]. The SSV2 requires more temporal reasoning [3, 28] while the K400 is relatively static biased [32, 8, 50].

**Fine-grained action recognition.** We evaluate the CAST on the fine-grained action recognition task: EPIC-KITCHENS-100 (EK100) [10]. In contrast to conventional action recognition, EK100 defines an action as a combination of a verb and a noun. Therefore, we refer to the action recognition in EK100 as *fine-grained action recognition*. Since fine-grained action recognition requires correctly predicting both the verb and the noun to recognize an action it is more challenging than conventional action recognition, which requires predicting a single action label: e.g., K400 or SSV2.

### 4.2 Implementation details

In this section, we briefly provide our experimental setup and implementation details. Please refer to the Appendix § B for complete implementation details. We conduct all the experiments with 16 NVIDIA GeForce RTX 3090 GPUs. We implement CAST using PyTorch and build upon the existing codebase of VideoMAE [56].

**Training.** We sample 16 frames from each video to construct an input clip. For the K400 dataset, we apply dense sampling [15], while for SSV2 and EK100, we use uniform sampling [61]. We then perform random cropping and resizing every frame into $224 \times 224$ pixels. We use the AdamW [39] optimizer with momentum betas of (0.9, 0.999) [7] and a weight decay of 0.05. By default, we train the model for 50 epochs, with the cosine annealing learning rate scheduling [38] and a warm-up period of 5 epochs. The default base learning rate, layer decay [2], and drop path are set to 0.001, 0.8, and 0.2, respectively. We freeze all the parameters of each expert, except for the B-CAST layer, adapters, and the last layer normalization. We set the batch size per GPU as 6 with update frequency of 2.

**Inference.** Given an input video, we randomly sample frames multiple times to construct input clips with multiple temporal views with multiple spatial crops. After the temporal frame sampling, we resize every frame so that the shorter side has 224 pixels. Then we perform spatial cropping to get multiple $224 \times 224$ crops for each clip. We get the final prediction by averaging the predictions on (temporal views) $\times$ (spatial crops). For the K400 dataset, we use (5 clips) $\times$ (3 crops) views, while for the other datasets, we use (2 clips) $\times$ (3 crops) views for the inference.

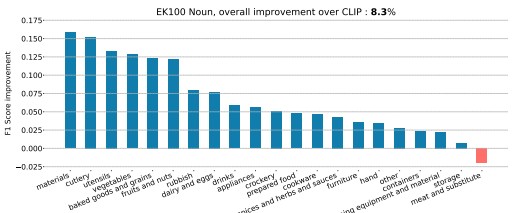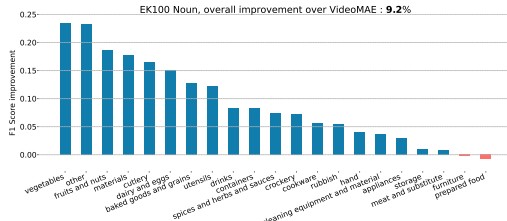

Figure 5: **Improvements of CAST over each expert on EK100 noun classes.** We show the super-category-wise weighted average F1 score improvement of CAST over each expert. (Left) Improvement over CLIP. CAST outperforms CLIP for every super-category except *meat and substitute*. (Right) Improvement over VideoMAE. CAST outperforms VideoMAE for every super-category except *furniture* and *prepared food*. Best viewed with zoom and color.

## 4.3 Balanced spatio-temporal understanding

In Figure 4 (a), we present the top-1 accuracies of several existing models. In the EK100 verb prediction task, VideoMAE outperforms ST-Adapter with a margin of 2.9 points (70.5% vs. 67.6%), while in the EK100 noun prediction task, ST-Adapter [42] outperforms VideoMAE [56] with a margin of 3.6 points (55.0% vs. 51.4%). As shown in Figure 4 (b), BEVT [62] outperforms AIM [72] with a margin of 2.5 points (70.6% vs. 68.1%) on the SSV2 dataset, while on the K400 dataset, AIM outperforms BEVT with a margin of 3.9 points (84.5% vs. 80.6%). We observe similar trends for other methods as well. Please refer to Section 4.6 for a detailed comparison. Our findings indicate that the performance of many existing models is significantly imbalanced toward either spatial or temporal understanding.

**Ingredients for balanced spatio-temporal understanding.** To achieve a more balanced spatio-temporal understanding, we can employ two expert models: a spatial expert and a temporal expert. For the spatial expert, we use CLIP [44], which has demonstrated impressive performance on various computer vision tasks. For the temporal expert, we use VideoMAE [56], which has shown favorable performance on temporal-biased tasks such as SSV2 and EK100 verb prediction tasks. (Please refer to Section 4.6 for the accuracy details.) While each expert is highly specialized in its own domain, we aim to create synergy between them by exchanging information to improve the balanced spatio-temporal understanding performance.

**Effect of CAST.** In Figure 4 (c), we gauge the balanced spatio-temporal understanding performance of our spatial expert, temporal expert, and CAST. For each method, we calculate the harmonic mean of top-1 accuracies for EK100 noun, EK100 verb, SSV2, and K400. The harmonic mean is an effective metric for gauging balanced performance because it gives more weight to lower-performing tasks. A higher harmonic mean value indicates that the performance over the different tasks is more balanced. Our spatial expert achieves an accuracy of 56.5%, while the temporal expert achieves an accuracy of 66.6%, and our CAST achieves an accuracy of 71.6%. These results validate the effectiveness of our proposed method, CAST, which allows our spatial and temporal experts to make synergistic predictions by exchanging information with each other through cross-attention.

## 4.4 Analysis on fine-grained action recognition

In this section, we provide a detailed analysis of how the proposed CAST improves the balanced spatio-temporal understanding in the fine-grained action recognition task: EK100.

**Category-level performance analysis.** In Figure 5, We present the EK100 noun super-category-wise weighted average F1 score improvement of CAST over our spatial expert (CLIP) and temporal expert (VideoMAE). In Figure 5 left, we observe that CAST significantly improves upon the spatial expert, CLIP, in several super-categories such as *cutlery*, *utensils*, and *vegetables*. These results indicate that the spatial expert achieves a more accurate understanding of fine-grained small objects interacting with the actors by leveraging the temporal context from the temporal expert. Similarly, in Figure 5 right, we observe that CAST significantly improves upon the temporal expert, VideoMAE, in several categories such as *vegetables* and *cutlery*. The trend is similar to the comparison with the spatial expert: CAST achieves more accurate understanding of fine-grained small objects by leveraging the fine-grained spatial context from CLIP.

**Qualitative analysis.** To better understand the effectiveness of CAST, we provide qualitative analysis on a few sample frames from the EK100 dataset in Figure 6. We show the predictions of CLIP, VideoMAE, and CAST. As expected, each expert model provides more accurate prediction in their respective tasks of expertise but shows weaker performance in the other task. In contrast,

Table 1: **Ablation study.** To validate the effect of each component, we show experimental results on the EPIC-Kitchens-100 dataset. In every experiment, we use the ViT-B/16 backbone for every expert. The best numbers are highlighted in gray.

(a) Effect of information exchange.

| Method | Verb | Noun | Act. |
|---|---|---|---|
| | \multicolumn Top-1 Acc. | | |
| Indep. experts w/o adapter | 70.7 | 50.1 | 40.0 |
| Indep. experts w/ adapter | 68.1 | 54.2 | 41.7 |
| Ensemble of experts w/ adapter | 68.2 | 55.3 | 42.9 |
| CAST | **72.5** | **60.3** | **48.7** |

(b) Different information exchange methods.

| Method | Late | Layer-wise | Verb | Noun | Act. |
|---|---|---|---|---|---|
| | | | \multicolumn Top-1 Acc. | | |
| Add | ✓ | | 68.9 | 56.6 | 44.2 |
| Concat | ✓ | | 69.2 | 56.4 | 44.5 |
| Lateral | | ✓ | 68.9 | 49.1 | 39.0 |
| CAST | | ✓ | **72.5** | **60.3** | **48.7** |

(c) B-CAST architecture.

| Method | Tune Param(M) | Verb | Noun | Act. |
|---|---|---|---|---|
| | | \multicolumn Top-1 Acc. | | |
| Identity | 18.1 | 68.1 | 54.2 | 41.7 |
| w/o adapter | 85.9 | 69.3 | 49.4 | 39.4 |
| X-attn.→adapter | 93.0 | 71.3 | 60.1 | 47.9 |
| B-CAST | 44.8 | **72.5** | **60.3** | **48.7** |

(d) Effect of projection ratio.

| Ratio | Verb | Noun | Act. |
|---|---|---|---|
| | \multicolumn Top-1 Acc. | | |
| 1/8 | 70.7 | 59.9 | 47.4 |
| 1/4 | 71.3 | 59.8 | 47.4 |
| 1/2 | **72.5** | **60.3** | **48.7** |
| 1 | 72.1 | 59.8 | 48.6 |

(e) Effect of cross-attention window shape.

| T2S | S2T | Verb | Noun | Act. |
|---|---|---|---|---|
| \multicolumn Window shape | | \multicolumn Top-1 Acc. | | |
| space-time | space-time | 71.0 | 59.3 | 47.2 |
| space-time | space | 71.9 | 60.3 | 48.4 |
| space | space | 72.3 | 60.2 | 48.5 |
| time | space | **72.5** | **60.3** | **48.7** |

(f) Effect of the number of cross-attention layers.

| 1-3 | 4-6 | 7-9 | 10-12 | Verb | Noun | Act. |
|---|---|---|---|---|---|---|
| \multicolumn X-attention layer | | | | \multicolumn Top-1 Acc. | | |
| | | | ✓ | 71.2 | 59.4 | 47.4 |
| | | ✓ | ✓ | 71.3 | 59.9 | 47.9 |
| | ✓ | ✓ | ✓ | 71.8 | 60.0 | 48.2 |
| ✓ | ✓ | ✓ | ✓ | **72.5** | **60.3** | **48.7** |

(g) Effect of bi-directional cross-attention.

| Method | Verb | Noun | Act. |
|---|---|---|---|
| | \multicolumn Top-1 Acc. | | |
| Indep. experts w/ adapter | 68.1 | 54.2 | 41.7 |
| S2T only | 71.2 | 55.0 | 43.7 |
| T2S only | 68.7 | 60.5 | 46.7 |
| CAST | **72.5** | **60.3** | **48.7** |

(h) Role of each expert.

| Spatial | Temporal | Verb | Noun | Act. |
|---|---|---|---|---|
| \multicolumn Expert | | \multicolumn Top-1 Acc. | | |
| CLIP | CLIP | 69.3 | 58.8 | 46.0 |
| VideoMAE | CLIP | 72.2 | 58.8 | 47.8 |
| VideoMAE | VideoMAE | 69.8 | 49.9 | 40.3 |
| CLIP | VideoMAE | **72.5** | **60.3** | **48.7** |

CAST consistently shows correct predictions for both noun and verb prediction tasks, such as *spoon* and *open*. The qualitative examples demonstrate the effectiveness of CAST in achieving balanced spatio-temporal understanding, which is essential for fine-grained action recognition.

## 4.5 Ablation study on CAST architecture

We conduct comprehensive ablation studies to examine the design choices for the proposed CAST architecture. Here we conduct all experiments on the EK100 [10] dataset with 16-frame input videos and report the top-1 accuracy on the validation set. We employ CLIP [44] as a spatial expert model and VideoMAE [56] as a temporal expert model. For a fair ablation study, we use the same hyperparameters for each experiment unless explicitly mentioned.

**Effect of information exchange.** We investigate whether CAST effectively achieves a synergistic effect by exchanging information between the two expert models. In Table 1 (a), we compare CAST with three baselines. i) A baseline using two independent expert models without any information exchange (fully fine-tuned). ii) The same baseline as i), but we add adapters and fine-tune the adapters and head only, iii) A test-time ensemble of two independent experts (with adapters and heads fine-tuning only). The baselines predict nouns using the spatial model and verbs using the temporal model. We observe that the two expert models using ensembling achieve an improvement in Action accuracy by at least 1.2 points compared to the baselines without any information exchange. Furthermore, CAST achieves a best Action accuracy of 48.7%. These results suggest that information exchange is crucial for achieving balanced spatio-temporal understanding.

**Comparison with simple information exchange baselines.** We compare CAST with simple information exchange baselines: i) late fusion with addition, ii) late fusion with concatenation, iii) layer-wise fusion using the bidirectional lateral connection (element-wise addition) with linear projection. We add adapters and fine-tune the adapters and head only in all three baselines, using our training recipe in Section 4.2 For the details of baseline fusion methods, please see Figure 7. We show the results in Table 1 (b). It is worth noting that both the late fusion and layer-wise lateral connection baselines result in a significant performance drop. Furthermore, we observe that layer-wise fusion without cross-attention yields inferior performance compared to the simple late fusion baselines. The results indicate that cross-attention in the bottleneck architecture is crucial for effective information exchange between spatial and temporal experts.

**Design of B-CAST module.** To explore the most effective and efficient way to integrate adapters for information exchange between the two expert models, we conduct an ablation study and present the results in Table 1 (c). For the details of the baselines, please see Figure 8. The first row of the table represents a baseline without the B-CAST module, which is equivalent to the identity function. Compared to this baseline, B-CAST achieves a significant improvement of 7.0 points in Action accuracy. The second row shows the performance of a baseline with cross-attention but without the

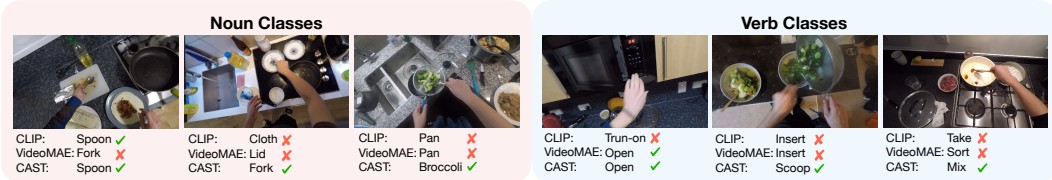

Figure 6: **Qualitative examples from EK100 comparing CLIP, VideoMAE, and the proposed CAST.** Each expert model shows more accurate predictions in their expertise but shows weaker performance on the other task. However, CAST consistently shows correct predictions for both tasks, demonstrating the effectiveness of the proposed spatio-temporal cross-attention mechanism.

bottleneck adapters. The 9.3-point gap between this baseline and B-CAST highlights the importance of bottleneck adapters for effective information exchange between the two expert models. The third row (*X-attn.→adapter*) is a baseline with the adapters after cross-attention. Compared to B-CAST, this baseline shows a 0.8 points drop in Action accuracy while having more than double the number of learnable parameters (44.9*M* vs. 93.0*M*). The results indicate that cross-attention in bottleneck is more effective and more efficient than the baseline. In summary, by placing cross-attention in the middle of the bottleneck adapter, B-CAST facilitates effective information exchange between the two experts and achieves a synergistic effect.

**Effect of projection ratio in bottleneck.**    In this study, we investigate the impact of the down projection ratio in the bottleneck architecture presented in Table 1 (d). The results demonstrate that a ratio of 1/2 yields the best performance. Notably, a ratio of 1 results in inferior performance, which we attribute to overfitting caused by the addition of more parameters.

**Effect of cross-attention window shape.**    We investigate the impact of the window shape in the cross-attention mechanism in the T2S and S2T modules in Table 1 (e). Please refer to Figure 3 (c) for the details of the window size. We maintain the same model capacity across different methods. Using space-time attention for both T2S and S2T modules results in the worst performance. We conjecture that learning joint space-time attention is challenging with the given model capacity [3]. On the other hand, using time attention in T2S and space attention in S2T yields the best performance. Consequently, we adopt this configuration throughout the paper.

**Effect of the number of cross-attention layers.**    We investigate the impact of the number of cross-attention layers used. To this end, we gradually increase the number of cross-attention layers starting from the top three layers, and report the results in Table 1 (f). As we increase the number of cross-attention layers, we observe a corresponding improvement in performance, as expected.

**Effect of bi-directional cross-attention.**    To validate the effectiveness of bi-directional information exchange, we ablate each cross attention at a time. We compare CAST with unidirectional information exchange baselines equipped with S2T or T2S cross-attention only. Each unidirectional information exchange baseline still has both experts. In Table 1 (g), compared to our CAST (48.7%), the S2T only baseline shows 5.0 points drop (43.7%) and the T2S only baseline shows 2.0 points drop (46.7%) in accuracy. The results validate the effectiveness of the proposed bi-directional cross-attention.

**Role of each expert.**    In Table 1 (h), we investigate the role of experts within CAST by controlling the assignment of models to each expert. We observe that we can achieve the best performance of 48.7% when we employ CLIP as our temporal expert and VideoMAE as our spatial expert. When we employ one VideoMAE as our spatial expert and another VideoMAE as our temporal expert, we obtain 40.3% accuracy. When we employ one CLIP as our spatial expert and another CLIP as our temporal expert, we obtain 46.0% accuracy.

Interestingly, when we revert the role of CLIP and VideoMAE, i.e., we employ VideoMAE as the spatial and CLIP as the temporal expert, we achieve a good performance of 47.8%. The results demonstrate that the B-CAST architecture facilitates effective information exchange between the two experts. Through the stacked B-CAST, the experts can learn high-quality spatio-temporal representations by exchanging information, even when the roles are reverted.

In summary, these findings suggest that CAST achieves optimal performance when models are assigned to expert roles that align with their strengths. CLIP serves as an effective spatial expert, whereas VideoMAE is more effective as a temporal expert. The B-CAST architecture encourages these experts to leverage their respective strengths through information exchange, resulting in enhanced spatio-temporal balanced understanding.

Table 2: **Comparison with the state-of-the-arts on the EK100, SSV2 and K400 datasets.** We show the Top-1 accuracy on each dataset and the harmonic mean (H.M.) of the Top-1 accuracies. The best performance is in **bold** and the second best is underscored.

| Method | GFLOPs/ View | EK100 Top-1 | | | SSV2 & K400 Top-1 | | | All |
| | | Verb | Noun | Act. | SSV2 | K400 | H.M. | H.M. |
|---|---|---|---|---|---|---|---|---|
| CLIP* [44] | 140 | 54.9 | 52.7 | 33.8 | 47.8 | 78.9 | 59.5 | 56.5 |
| EVL [35] | 592 | - | - | - | 62.4 | 82.9 | 71.2 | - |
| ST-Adapter [42] | 607 | 67.6 | 55.0 | - | 69.5 | 82.7 | 75.5 | 67.3 |
| AIM [72] | 404 | 64.8 | 55.5 | 41.3* | 68.1 | 84.5 | 75.4 | 66.7 |
| MBT [40] | 936 | 64.8 | 58.0 | 43.4 | - | 80.8 | - | - |
| ViViT FE [1] | 990 | 66.4 | 56.8 | 44.0 | 65.9 | 81.7 | 73.0 | 66.6 |
| TimeSformer [3] | 2380 | - | - | - | 62.4 | 80.7 | 70.4 | - |
| MViT [12] | 170 | - | - | - | 67.7 | 80.2 | 73.4 | - |
| MFormer [43] | 1185 | 67.1 | 57.6 | 44.1 | 68.1 | 80.2 | 73.7 | 67.3 |
| ORViT MF [21] | - | 68.4 | 58.7 | 45.7 | 67.9 | - | - | - |
| Video Swin [37] | 282 | - | - | - | 69.6 | 82.7 | 75.8 | - |
| BEVT [62] | 282 | - | - | - | 70.6 | 80.6 | 75.3 | - |
| VideoMAE [56] | 180 | 70.5 | 51.4 | 41.7* | 70.8 | 81.5 | 75.8 | 66.6 |
| MeMViT [68] | 59 | 70.6 | 58.5 | 46.2 | - | - | - | - |
| OMNIVORE [16] | - | 69.5 | 61.7 | **49.9** | 71.4 | 84.0 | 77.2 | 70.8 |
| MTV-HR [71] | 930 | 68.0 | 63.1 | 48.6 | 68.5 | 82.4 | 74.8 | 69.8 |
| CAST | 391 | 72.5 | 60.9 | 49.3 | 71.6 | 85.3 | **77.9** | **71.6** |

*We conduct experiments with our own implementation.

## 4.6 Comparison with state-of-the-art

In this section, we evaluate the performance of CAST and state-of-the-art methods in terms of balanced spatio-temporal understanding on multiple datasets, as shown in Table 2. For each method, in addition to reporting the top-1 accuracy of each task, we report the harmonic mean of top-1 accuracies for i) SSV2, and K400, and ii ) EK100 verb, EK100 noun, SSV2, and K400. For comparison with state-of-the-art models, we have set different hyperparameters than those used in our ablation study. Please refer to Table 4 for the details. For fair comparisons of the computation complexity, we show the GFLOPs/View. In cases where a compared method shows various GFLOPs/View depending on the dataset, we specifically note the lowest GFLOPs/View value for reference. For more detailed comparison of computation complexity, please refer to the Appendix § E.

We observe that among the CLIP-based methods (the second group in Table 2), AIM [72] achieves favorable performance on the static-biased K400 dataset, with 84.5% accuracy. However, AIM shows a relatively lower performance of 68.1% on the temporal-biased SSV2. On the other hand, VideoMAE [56], one of the state-of-the-art methods, shows 70.8% accuracy on the SSV2 dataset, which is more competitive than AIM. However, VideoMAE shows a lower accuracy of 81.5% on the K400 dataset, less competitive than AIM. Our proposed method, CAST, demonstrates favorable performance on both the SSV2 (71.6%) and K400 (85.3%) datasets, resulting in a harmonic mean of 77.9%, which is higher than that of AIM (75.4%) and VideoMAE (75.8%). CAST shows a more balanced spatio-temporal understanding than the existing methods. Additionally, CAST shows favorable performance in fine-grained action recognition on the EK100 dataset. CAST achieves a competitive Action accuracy of 49.3%, which is the second best among the compared methods.

In terms of the overall harmonic mean of EK100 verb, EK100 noun, SSV2, and K400 accuracies, CAST shows the best performance of 71.6%. The results highlight the effectiveness of CAST. By exchanging information between spatial and temporal experts, our CAST shows a favorable balanced spatio-temporal understanding performance.

## 5 Conclusions

In this paper, we present a solution to the problem of action recognition models lacking a balanced spatio-temporal understanding of videos. The proposed method, CAST, incorporates a spatial expert and a temporal expert that exchange information through cross-attention to achieve synergistic predictions. Our extensive experiments on datasets with varying characteristics demonstrate that CAST outperforms both individual expert models and existing methods in terms of a balanced spatio-temporal understanding measure: the harmonic mean of accuracies on the datasets. The results highlight the effectiveness of CAST in achieving a balanced spatio-temporal understanding of videos, and suggest that CAST could have broad applicability in the field of video understanding.

**Acknowledgment.** This work was partly supported by Institute of Information & communications Technology Planning & Evaluation (IITP) grant funded by the Korea government(MSIT) (No.RS-2022-00155911, Artificial Intelligence Convergence Innovation Human Resources Development (Kyung Hee University)); by the Institute of Information & Communications Technology Planning & Evaluation (IITP) grant funded by the Korea Government (MSIT) (Artificial Intelligence Innovation Hub) under Grant 2021-0-02068; by the National Research Foundation of Korea(NRF) grant funded by the Korea government(MSIT) (No. 2022R1F1A1070997).

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

# Appendix

In this appendix, we provide additional architecture/implementation/dataset details, experimental settings, quantitative/qualitative results, limitations of our method, and broader impact of our method to complement the main paper. We organize the appendix as follows:

- A. Architecture details of our framework
- B. Implementation details and experimental settings
- C. Details of datasets in our experiments
- D. Additional quantitative and qualitative results
- E. Comparison with State-of-the-Art with additional information
- F. Class-wise F1 score on EK100 verb and noun classes
- G. Additional qualitative analysis on EK100
- H. Limitations
- I. Broader impacts

# A    Architecture Details

In this section, we provide details of our B-CAST architecture. Let us assume we employ CLIP [44] as a spatial expert and VideoMAE [56] as a temporal expert.

Table 3: **Stage-wise details of the two experts in B-CAST.** We provide a detailed description of each operation performed in each expert. The input for this example is a video consisting of 16 frames. MHCA represents multi-head cross-attention applied with a specific window shape: either time-attention or space-attention. In the description, B, D, T, and N represent the batch size, embedding dimension, temporal sequence length, and spatial sequence length, respectively. We omit the Layer Normalization and activation functions for simplicity.

| B-CAST Stage | Spatial Expert | | Temporal Expert | |
|---|---|---|---|---|
| | **Remark** | **Output Tensor Shape** | **Remark** | **Output Tensor Shape** |
| Up Projection | Linear projection *with ratio = 2.0* | $\mathbf{Y_s}$:(196+1)×B·8×768 | Linear projection *with ratio = 2.0* | $\mathbf{Y_t}$:B×8·196×768 |
| Post Processing | Attach CLS token of $\mathbf{Y_s}$ Reshape: N×B·T×D | $\mathbf{Y_s}$:(196+1)×B·8×384 | Reshape: B×T·N×D | $\mathbf{Y_t}$:B×8·196×384 |
| Cross-Attention | T2S MHCA($\mathbf{Y}_s$, $\mathbf{Y}_t$) Window shape: *time* | $\mathbf{Y}_s$:B·196×8×384 | S2T MHCA($\mathbf{Y}_t$, $\mathbf{Y}_s$) Window shape: *space* | $\mathbf{Y}_t$:B·8×196×384 |
| Positional Embeddings | # parameters: 8×384 | $\mathbf{Y}_s$:B·196×8×384 $\mathbf{Y}_t$:B·196×8×384 | # parameters: 196×384 | $\mathbf{Y}_t$:B·8×196×384 $\mathbf{Y}_s$:B·8×196×384 |
| Pre processing | Detach CLS token of $\mathbf{Y_s}$ Reshape:B·N×T×D | $\mathbf{Y}_s$:B·196×8×384 $\mathbf{Y}_t$:B·196×8×384 | Detach CLS token of $\mathbf{Y_s}$ Reshape: B·T×N×D | $\mathbf{Y}_t$:B·8×196×384 $\mathbf{Y}_s$:B·8×196×384 |
| Gather Features | Gather $\mathbf{Y}_t$ from *Temporal Expert* | $\mathbf{Y_s}$:(196+1)×B·8×384 $\mathbf{Y}_t$:B×8·196×384 | Gather $\mathbf{Y}_s$ from *Spatial Expert* | $\mathbf{Y_t}$:B×8·196×384 $\mathbf{Y_s}$:(196+1)×B·8×384 |
| Down Projection | Linear projection *with ratio = 0.5* | $\mathbf{Y_s}$:(196+1)×B·8×384 | Linear projection *with ratio = 0.5* | $\mathbf{Y_t}$:B×8·196×384 |
| Input of B-CAST | - | $\mathbf{Y_s}$:(196+1)×B·8×768 | - | $\mathbf{Y_t}$:B×8·196×768 |

**B-CAST architecture.** In Table 3, we provide stage-wise details of the two experts in B-CAST. Given an input from multi-head self-attention (MHSA) layer, we first pass it through the linear projection layer of each expert. Subsequently, the two experts exchanges their features each other. The multi-head cross-attention (MHCA) layer of each expert enables the effectively exchange of information between the two experts. Afterward, we pass the output tensors from the B-CAST

through the Feed-Forward Network (FFN). We repeat this process in a stacked manner, consisting of 12 blocks, each comprising the MHSA module along with adapters, B-CAST, and FFN module along with adapters. Finally, we feed the resulting tensors to a classification head to predict action.

**Architecture details of different information exchange methods.** In Figure 7, we visualize the architectures of the simple information exchange baselines presented in Table 1 (b) of the main paper. These baselines involve fully fine-tuning the two expert models without the use of adapters, following the training recipe outlined in Appendix § B. In Figure 7 (a), we present the add baseline, which facilitates information exchange between the spatial and temporal experts through late fusion using element-wise addition. In Figure 7 (b), we present the concat baseline, which exchanges the information between the spatial and the temporal experts through late fusion using concatenation. In Figure 7 (c), we show the lateral connection baseline, which enables information exchange between the spatial and temporal experts through layer-wise lateral connections, incorporating linear projections.

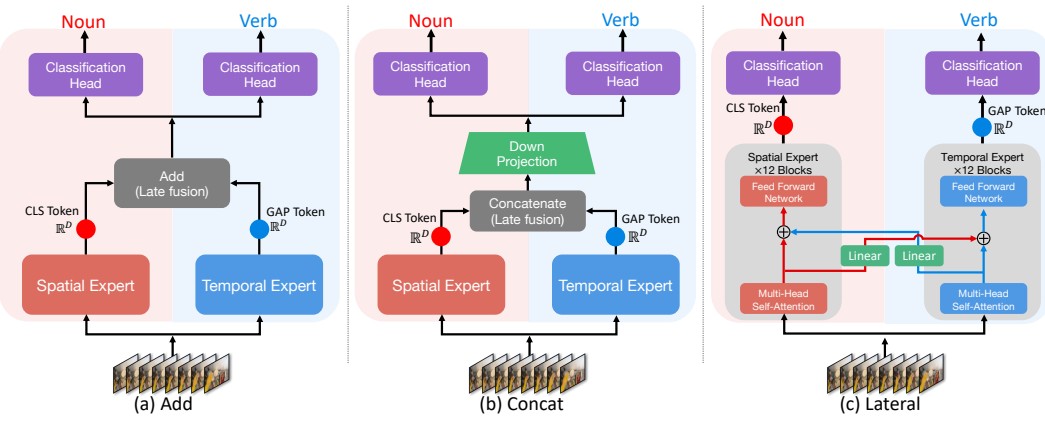

Figure 7: **Architecture visualization of the different information exchange baselines.**

**Architecture details of the baselines in the B-CAST architecture ablation study.** In Figure 8, we visualize the architectures of the baselines used in the B-CAST architecture ablation study, as presented in Table 1 (c) of the main paper. In Figure 8 (a), we illustrate the identity baseline, which serves as a baseline without the B-CAST module. In Figure 8 (b), we present the w/o adapter baseline, which includes cross-attention but does not incorporate the bottleneck adapters. In Figure 8 (c), we show the X-attn.→adapter baseline, which uses adapters positioned after the cross-attention stage. For reference, we include our B-CAST architecture in Figure 8 (d), which represents the final model used in our study.

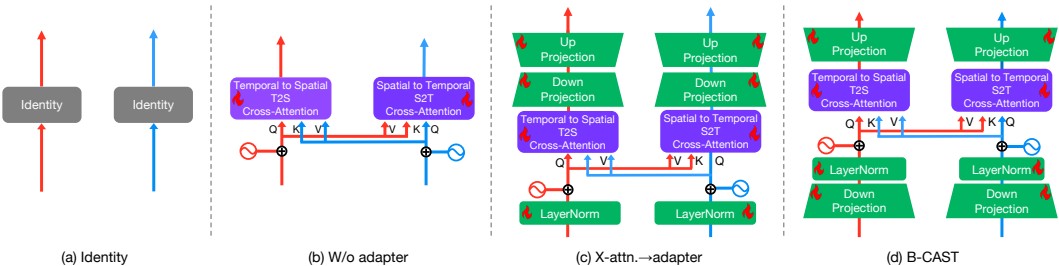

Figure 8: **Architecture visualization of the baselines used in the B-CAST ablation study.**

**Classification head.** We use different classification strategies for conventional action recognition and fine-grained action recognition as shown in Figure 9. i) For conventional action recognition datasets, i.e., Kinetics-400 (K400) and Something-Something-V2 (SSV2), CAST combines the CLS and GAP tokens from the two experts to predict actions, as depicted in Figure 9 (a). The spatial expert, CLIP [44], generates one CLS token for each frame. To obtain a single CLS token,

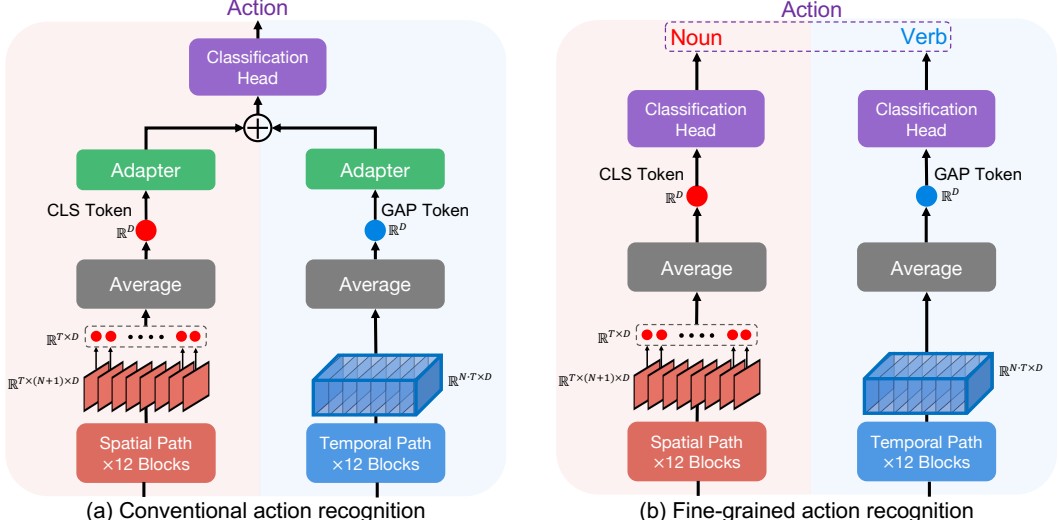

Figure 9: **Classification head architecture.** We use different classification strategies for conventional action recognition shown in (a) and fine-grained action recognition shown in (b). $T$, $N$, and $D$ denote the temporal sequence length, spatial sequence length, and embedding dimension respectively. The output from the spatial expert consists of frame-level feature vectors. Each frame-level feature vector consists of $N$ patches and one CLS token. The output of the temporal expert is a video-level feature vector, consisting of $N \cdot T$ patches.

we take the average of the CLS tokens from all frames of the input video. The temporal expert, VideoMAE [56], performs global average pooling on all output features to obtain a single GAP token. After passing through adapters, we add the CLS and GAP tokens together and feed the token into a linear classification head. ii) In a fine-grained action recognition dataset, i.e., EPIC-KITCHENS-100 (EK100), where a model needs to predict both verb and noun, we use two separate classification heads instead of a single head shown in Figure 9 (b). Specifically, we feed the CLS token from the spatial expert into a linear classification head for noun prediction and the GAP token from the temporal expert into another linear classification head for verb prediction.

## B   Implementation Details

In this section, we provide more details of our experimental setup and implementation of each dataset. We conduct the experiments with 16 NVIDA GeForce RTX 3090 GPUs. We implement CAST using PyTorch and build upon the existing codebase of VideoMAE [56].

**Data preprocessing.**   After sampling the videos to 16 frames, We randomly crop each frame of the video and resize it to 224×224. We also apply data augmentation techniques, including mixup [73], label smoothing [55], horizontal flip, color jitter, and randaugment [9], repeated augmentation [22] to diversify the training data. We do not use horizontal flip on SSV2. Note that if the patch embedding layer of the temporal expert has a time stride value of 2, e.g., VideoMAE [56], we only take even frames for the spatial pathway. After the patch embedding layers, both experts take an input of 3 channels × 8 frames × 224 width × 224 height. We use the same data preprocessing protocol in all the experiments.

**Model training.**   We conduct experiments using 2 nodes, each equipped with 8 GPUs. To ensure efficient multi-node training, we utilize the DeepSpeed [3] library. Additionally, we increase the effective batch size by implementing gradient accumulation to update the model weights. For the EK100 dataset, we set the update frequency to 4 iterations [4], resulting in a total batch size of 24

---

[3]https://github.com/microsoft/DeepSpeed

[4]We use the update frequency of 2 iterations for the additional quantitative analysis experiments in Appendix § D.

Table 4: **Hyperparameters used and the fine-tuning configuration for each dataset.**

| Config | EK100 | K400 | SSV2 |
|---|---|---|---|
| Optimizer | AdamW [39] | AdamW | AdamW |
| Base learning rate | 1e-3 | 1e-3 | 1e-3 |
| Weight decay | 0.05 | 0.05 | 0.05 |
| Optimizer momentum [7] | $\beta_1, \beta_2 = 0.9, 0.999$ | $\beta_1, \beta_2 = 0.9, 0.999$ | $\beta_1, \beta_2 = 0.9, 0.999$ |
| Gpu per batch size | 6 | 6 | 6 |
| Update frequency | 4 | 6 | 4 |
| Learning rate schedule | cosine decay [38] | cosine decay | Cosine decay |
| Warmup epochs | 5 | 5 | 5 |
| Training epochs | 50 | 70 | 50 |
| Flip augmentation | *yes* | *yes* | *no* |
| Color jitter | 0.4 | 0.4 | 0.4 |
| RandAug [9] | (9, 0.5) | (9, 0.5) | (9, 0.5) |
| Label smoothing [55] | 0.1 | 0.1 | 0.1 |
| Mixup [73] | 0.8 | 0.8 | 0.8 |
| Drop path | 0.2 | 0.2 | 0.3 |
| Layer-wise lr decay [2] | 0.75 | 0.75 | 0.75 |

per GPU. We linearly scale the base learning rate, then *actual lr = base lr × total batch size*/256. In Table 4, we summarize the fine-tuning configuration and the hyperparameters used in Table 2, Table 9, Table 10, and Table 11.

**Pre-trained weights.** We take the off-the-shelf pre-trained weights of the two expert models. For our main temporal expert, we take the VideoMAE [56] weights pre-trained on the K400, and SSV2 datasets from the official repository[5]. Since VideoMAE does not provide pre-trained weights specifically for the EK100, we pre-train VideoMAE on the EK100 without incorporating extra video datasets. The pre-training process follows the recipe described in the VideoMAE paper [56]. For all other experiments, we make use of the pre-trained weights provided by the respective model repositories to ensure consistency and reliability in our results.

## C   Datasets.

**Action recognition** We evaluate our B-CAST module on two video datasets:Kinetics400 (K400) [24], Something-Something-V2 (SSV2) [19]. i) K400 is a large-scale third-person video dataset for action recognition that contains around 300K video clips and 400 human action classes. The dataset is split into train/val/test, with 240K/20K/40K video clips. The videos are all trimmed to around 10 seconds from different YouTube video. ii) SSV2 contains over 220K short video clips labeled video clips of humans performing pre-defined, basic actions with everyday objects. The dataset is split into train/val/test, with 168K/24K/27K and have 174 human-objects interaction categories.

**Fine-grained action recognition** We evaluate our B-CAST module on a Compositional Action dataset: EPIC-KITCHENS-100 (EK100) [10]. EK100 is a large-scale(100hours) egocentric video dataset that records several days of kitchen unscripted activities. It consists of 90K action segments, which are split into train/val/test sets of 67K/10K/13K. Differ from preceding two datasets, EK100 define an action as a combination of a verb and a noun. Because it requires matching both verbs and nouns to recognize an action, it is more challenging than recognizing actions in a dataset where actions are represented by a single label e.g., Kinetics-400, Something-Something-V2.

## D   Additional Quantitative Analysis

In this section, we provide additional results to complement the main paper. We demonstrate (1) the generality of CAST with different ViT architectures and pre-trained weights in Appendix § D.1, and (2) the effect of B-CAST-specific positional embeddings in Appendix § D.2.

---

[5]https://github.com/MCG-NJU/VideoMAE

## D.1 Generality

In this section, we showcase the generality of the proposed method. We demonstrate that CAST works well with any ViT backbone and pre-trained weights. We conduct all the experiments on the EK100 dataset.

**CAST is pre-training dataset agnostic.** CAST is pre-training dataset agnostic. In Table 5, we compare CAST to a CAST variant where we replace CLIP with a ViT-B model pre-trained on the ImageNet-21K (IN21K) dataset. As a reference, we also show the performance of the independent experts using two separate expert models without any information exchange. When equipped with IN21K pre-trained ViT-B, CAST still achieves reasonable performance with an Action accuracy of 45.5%, while the baseline of independent experts achieves 40.4% accuracy.

Table 5: **Effect of CLIP pre-trained weights.**

| Method | Pre-training dataset | | Information | Top-1 Acc. | | |
| --- | --- | --- | --- | --- | --- | --- |
| | Spatial expert | Temporal expert | Exchange | Verb | Noun | Action |
| Independent experts | IN21K | EK100 | ✗ | 69.7 | 50.9 | 40.4 |
| CAST | IN21K | EK100 | ✓ | 70.9 | 56.8 | 45.5 |
| CAST | CLIP | EK100 | ✓ | 72.5 | 60.3 | 48.7 |

In Table 6, we analyze the effect of pre-training datasets on the temporal expert, VideoMAE, in CAST. We show the results of using EK100, SSV2, and K400 pre-trained VideoMAE weights. We also investigate the impact of pre-training datasets on the spatial expert, CLIP. We present the results of using IN21K and CLIP pre-trained weights. When using CLIP pre-trained weights for the spatial expert, we observe stable Action accuracy ranging from a minimum of 48.7% to a maximum of 49.4%. We observe a similar trend when we use IN21K pre-trained CLIP weights. These results demonstrate that the proposed method is agnostic to the pre-training datasets.

Table 6: **Effect of pre-trained weights.**

| Pre-training dataset | | Top-1 Acc. | | |
| --- | --- | --- | --- | --- |
| Spatial expert | Temporal expert | Verb | Noun | Act. |
| IN-21K | EK100 | 70.9 | 56.8 | 45.5 |
| IN-21K | SSV2 | 71.6 | 56.1 | 45.3 |
| IN-21K | K400 | 72.2 | 56.4 | 45.9 |
| CLIP | EK100 | 72.5 | 60.3 | 48.7 |
| CLIP | SSV2 | 73.3 | 60.0 | 49.0 |
| CLIP | K400 | 72.9 | 60.4 | 49.4 |

**CAST is model-agnostic.** In Table 7, we demonstrate that CAST is model-agnostic. In this experiment, we replace our spatial and temporal expert models with other existing models. We employ EVA [53], an extension of CLIP that has shown excellent performance in various vision tasks, as our spatial expert. We employ MVD [63] as our temporal expert. The results show that CAST achieves similar performance when we employ different models as the experts. For example, EVA + MVD achieves an Action accuracy of 49.2%, while CLIP + VideoMAE achieves 48.7% Action accuracy.

Table 7: **Effect of employing different models as experts.**

| Model architecture | | Top-1 Acc. | | |
| --- | --- | --- | --- | --- |
| Spatial expert | Temporal expert | Verb | Noun | Act. |
| CLIP | VideoMAE | 72.5 | 60.3 | 48.7 |
| CLIP | MVD | 73.1 | 60.1 | 49.3 |
| EVA | VideoMAE | 73.1 | 59.8 | 49.1 |
| EVA | MVD | 73.7 | 60.1 | 49.2 |

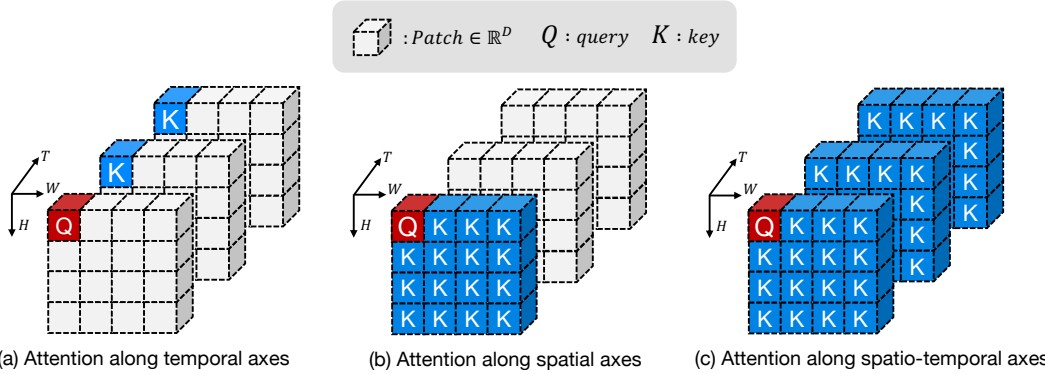

(a) Attention along temporal axes      (b) Attention along spatial axes      (c) Attention along spatio-temporal axes

Figure 10: **Visualization of attention window shape.**

## D.2  B-CAST-specific positional embeddings

We investigate the impact of B-CAST-specific positional embeddings, as depicted in Figure 8 (d). In the temporal-to-spatial (T2S) cross-attention, the spatial expert attends along the temporal axis, as shown in Figure 10 (a), while in the self-attention stage during pre-training, the spatial expert attends along the spatial axes, as depicted in Figure 10 (b). Consequently, the spatial expert lacks information about temporal patch sequences. Similarly, in the spatial-to-temporal (S2T) cross-attention, the temporal expert attends along the spatial axis, as illustrated in Figure 10 (b), while during self-attention stage during pre-training, the temporal expert attends along the spatio-temporal axes, as shown in Figure 10 (c). Consequently, the temporal expert lacks information about spatial-only patch sequences. To address this limitation, we introduce new learnable positional embeddings that are specific to T2S and S2T cross-attention.

As shown in Table 8, adding the B-CAST-specific positional embeddings boost the performance by 1.2 points compared to without using the B-CAST-specific positional embeddings. The results indicate the effectiveness of the B-CAST-specific positional embeddings.

Table 8: **Effect of B-CAST-specific positional embeddings.**

|  | Top-1 Acc. | | |
| --- | --- | --- | --- |
| Method | Verb | Noun | Action |
| CAST w/o B-CAST-specific positional embeddings | 71.3 | 59.7 | 47.5 |
| CAST w/ B-CAST-specific positional embeddings | 72.5 | 60.3 | 48.7 |

## E  Comparison with State-of-the-Art

To provide more comprehensive information, we augment the tables for comparison with state-of-the-art in the main paper. Table 9, Table 10, and Table 11, corresponding to the respective datasets, include additional details such as the number of frames per clip, the number of temporal and spatial views used for inference, the computation complexity, and the number of learnable parameters for each model. For the details of the hyperparameters used, please refer to Table 4.

## F  Class-Wise Performance Comparison

We provide class-wise F1 score improvement of CAST over our spatial expert (CLIP) and our temporal expert (VideoMAE). We show the verb-class-wise F1 score improvement over CLIP in Figure 11 and the improvement over VideoMAE in Figure 12. We show the noun-class-wise F1 score improvement over CLIP in Figure 13 and the improvement over VideoMAE in Figure 14.

Table 9: **Comparison with the state-of-the-arts on the EPIC-Kitchens-100 dataset.** We show the Top-1 accuracy for Action, Noun, and Verb prediction tasks as well as the number of frames per clip, the number of temporal and spatial views used for inference, the computation complexity, and the number of learnable parameters for each model. The best performance is in **bold** and the second best is underscored.

| Method | Backbone | Frames | Views | TFLOPs | Learnable Param (M) | Verb | Noun | Action |
|---|---|---|---|---|---|---|---|---|
| SlowFast [15] | ResNet50 | - | - | - | - | 54.9 | 50.0 | 38.5 |
| GSF [52] | ResNet50 | 16 | 3×2 | 0.4 | - | 68.8 | 52.7 | 44.0 |
| MoViNet [27] | MoViNet-A6 | - | - | - | 31 | 72.2 | 57.3 | 47.7 |
| CLIP* [44] | ViT-B | 8 | 2×3 | 0.84 | 86 | 55.5 | 52.3 | 33.9 |
| AIM* [72] | ViT-B | 16 | 2×3 | 2.42 | 14 | 64.8 | 55.5 | 41.3 |
| ST-Adapter [42] | ViT-B | 8 | 3×1 | - | - | 67.6 | 55.0 | - |
| MBT [40] | ViT-B | 32 | - | - | - | 64.8 | 58.0 | 43.4 |
| ViViT FE [1] | ViT-L | 32 | 4×1 | 15.92 | 311 | 66.4 | 56.8 | 44.0 |
| MFormer [43] | ViT-L | 32 | 3×1 | 3.56 | - | 67.1 | 57.6 | 44.1 |
| ORViT-MF-HR [21] | ViT-B | 16 | 10×3 | - | - | 68.4 | 58.7 | 45.7 |
| VideoSwin [37] | Swin-B | - | - | - | 89 | 67.8 | 57.0 | 46.1 |
| VideoMAE* [56] | ViT-B | 16 | 2×3 | 1.08 | 87 | 70.5 | 51.4 | 41.7 |
| MeMViT [68] | ViT-B | 16 | 1×1 | 0.06 | - | 70.6 | 58.5 | 46.2 |
| OMNIVORE [16] | Swin-B | 32 | - | - | - | 69.5 | 61.7 | 49.9 |
| MTV-HR [71] | MTV-B | 32 | 4×1 | 3.72 | 310 | 68.0 | 63.1 | 48.6 |
| CAST w/ CLIP & VideoMAE pretrained on EPIC-KITCHENS-100 | CAST-B | 16 | 2×3 | 2.35 | 45 | 72.5 | 60.9 | 49.3 |

*We conduct experiments with our own implementation.

Table 10: **Comparison with the state-of-the-arts on the Something-Something-V2 dataset.** We show the Top-1 accuracy as well as the number of frames per clip, the number of temporal and spatial views used for inference, the computation complexity, and the number of learnable parameters for each model. The best performance is in **bold** and the second best is underscored.

| Method | Backbone | Frames | Views | TFLOPs | Learnable Param (M) | Top-1 Acc. |
|---|---|---|---|---|---|---|
| SlowFast [15] | ResNet101 | 8+32 | 1×3 | 0.32 | - | 63.1 |
| CLIP* [44] | ViT-B | 8 | 2×3 | 0.84 | - | 43.2 |
| EVL [35] | ViT-B | 32 | 1×3 | 2.05 | 29 | 62.4 |
| ST-Adapter [42] | ViT-B | 32 | 3×1 | 1.96 | - | 69.5 |
| AIM [72] | ViT-B | 32 | 1×3 | 2.50 | 14 | 69.1 |
| ViViT FE [1] | ViT-L | 32 | 4×3 | 11.89 | 311 | 65.9 |
| TimeSformer [3] | ViT-L | 64 | 1×3 | 7.14 | - | 62.4 |
| MViT [12] | ViT-B | 64 | 1×3 | 1.37 | 37 | 67.7 |
| MFormer [43] | ViT-L | 32 | 1×3 | 3.56 | - | 68.1 |
| Video Swin [37] | Swin-B | 32 | 1×3 | 0.96 | 89 | 69.6 |
| BEVT [62] | Swin-B | 32 | 1×3 | 0.96 | - | 70.6 |
| VideoMAE [56] | ViT-B | 16 | 2×3 | 1.08 | 87 | 70.8 |
| ORViT-MF-L [21] | ViT-L | 32 | 1×3 | - | - | 69.5 |
| OMNIVORE [16] | Swin-B | 32 | - | - | - | 71.4 |
| MTV-HR [71] | MTV-B | 32 | 4×3 | 11.16 | 310 | 68.5 |
| CAST w/ VideoMAE pretrained on Something-Something-V2 | CAST-B | 16 | 2×3 | 2.35 | 45 | 71.6 |

*We conduct experiments with our own implementation.

# G Qualitative Analysis

To better understand the effectiveness of CAST, we provide qualitative analysis on more sample frames from the EK100 dataset in Figure 15. Each expert model provides more accurate prediction in their respective tasks of expertise but shows weaker performance in the other task. In contrast, CAST consistently shows correct predictions for both noun and verb prediction tasks. The qualitative examples demonstrate the effectiveness of CAST in achieving balanced spatio-temporal understanding, which is essential for fine-grained action recognition.

# H Limitations

Despite achieving a good balanced spatio-temporal understanding performance, CAST has a few limitations as well. CAST has a small number of learnable parameters since we freeze the expert models except for the adapters. However, the computational complexity of CAST is not negligible due to the utilization of two expert models. Due to resource limitations, we are unable to conduct experiments on various input video lengths and model sizes. Lastly, cross-attention layers require features of the same dimension for attention operation. Therefore, if the two model have significantly different architectures, it might be challenging for CAST to employ the two models.

Table 11: **Comparison with the state-of-the-arts on the Kinetics400 dataset.** We show the Top-1 accuracy as well as the number of frames per clip, the number of temporal and spatial views used for inference, the computation complexity, and the number of learnable parameters for each model. The best performance is in **bold** and the second best is underscored.

| Method | Backbone | Frames | Views | TFLOPs | Learnable Param (M) | Top-1 Acc. |
|---|---|---|---|---|---|---|
| SlowFast [15] | ResNet101 | 80 | 10×3 | 7.02 | - | 79.8 |
| X3D [13] | X3D-XL | 16 | 10×3 | 1.45 | - | 79.1 |
| MoViNet [27] | MoViNet-A6 | 120 | 1×1 | 0.39 | - | 81.5 |
| UniFormer [31] | Hybrid-B | 32 | 4×3 | 3.12 | 50 | 83.0 |
| CLIP* [44] | ViT-B | 8 | 5×3 | 2.2 | 86 | 77.3 |
| EVL [35] | ViT-B | 32 | 3×1 | 1.78 | 29 | 84.2 |
| ST-Adapter [42] | ViT-B | 32 | 3×1 | 1.82 | - | 82.7 |
| Text4Vis [69] | ViT-B | 16 | 4×3 | - | - | 83.6 |
| AIM [72] | ViT-B | 32 | 3×1 | 2.43 | 11 | 84.7 |
| X-CLIP [41] | ViT-B | 16 | 4×3 | 3.44 | - | 84.7 |
| ViViT FE [1] | ViT-L | 128 | 1×3 | 11.94 | 311 | 81.7 |
| TimeSformer [3] | ViT-L | 96 | 1×3 | 25.06 | 430 | 80.7 |
| MViT [12] | ViT-B | 32 | 5×1 | 0.85 | 37 | 80.2 |
| BEVT [62] | Swin-B | 32 | 4×3 | 3.38 | 88 | 80.6 |
| MFormer [43] | ViT-L | 32 | 10×3 | 35.55 | - | 80.2 |
| Video Swin [37] | Swin-L | 32 | 4×3 | 7.25 | 197 | 83.1 |
| VideoMAE [56] | ViT-B | 16 | 5×3 | 2.7 | 87 | 81.5 |
| OMNIVORE [16] | Swin-B | 32 | - | - | - | 84.0 |
| MTV-HR [71] | MTV-B | 32 | 4×3 | 11.16 | 310 | 82.4 |
| CAST w/ VideoMAE pretrained on Kinetics-400 | CAST-B | 16 | 5×3 | 5.87 | 45 | **85.3** |

*We conduct experiments with our own implementation.

# I  Broader Impacts

Our work is on the task of human action recognition from videos. Surveillance could be one application, which might have privacy related concerns when the technology is deployed. Other consumer applications like personal or internet video search and tagging is expected to benefit individuals and organizations alike by helping them more efficiently maintain human centered data.

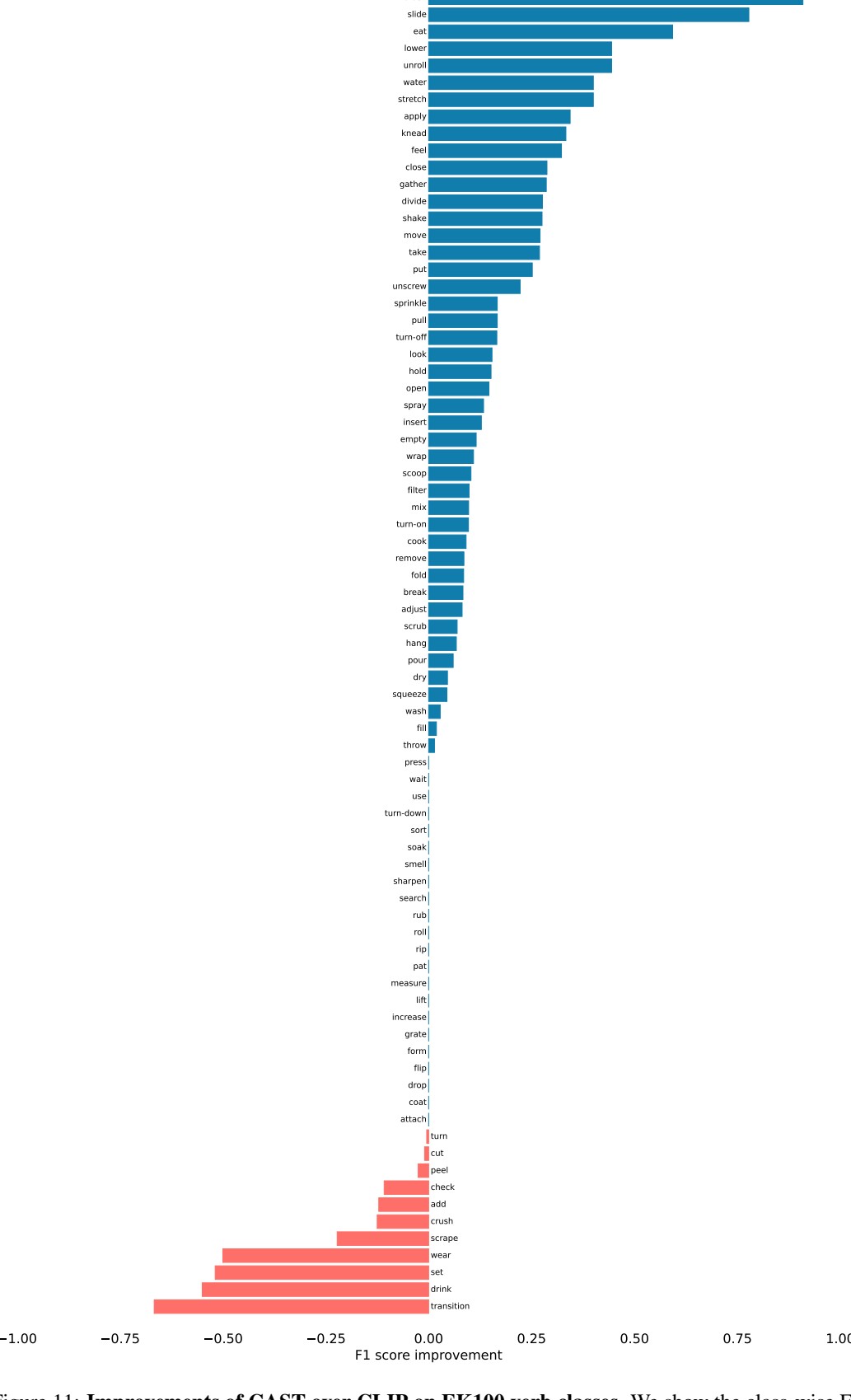

Figure 11: **Improvements of CAST over CLIP on EK100 verb classes.** We show the class-wise F1 score improvement of CAST over CLIP. CAST achieves an improvement of 17.8 points on average. Best viewed with zoom and color.

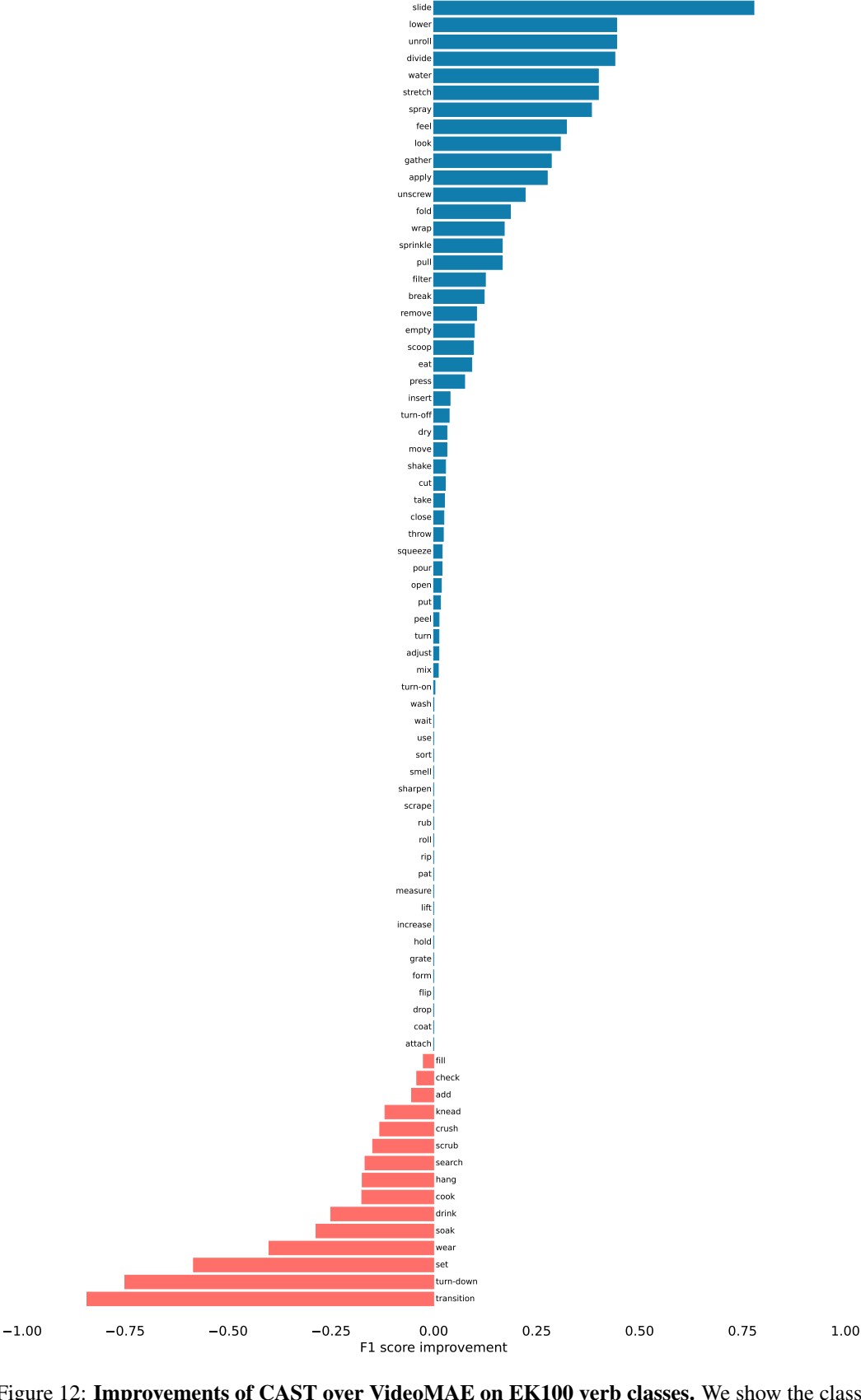

Figure 12: **Improvements of CAST over VideoMAE on EK100 verb classes.** We show the class-wise F1 score improvement of CAST over VideoMAE. CAST achieves an improvement of 2.2 points in on average. Best viewed with zoom and color.

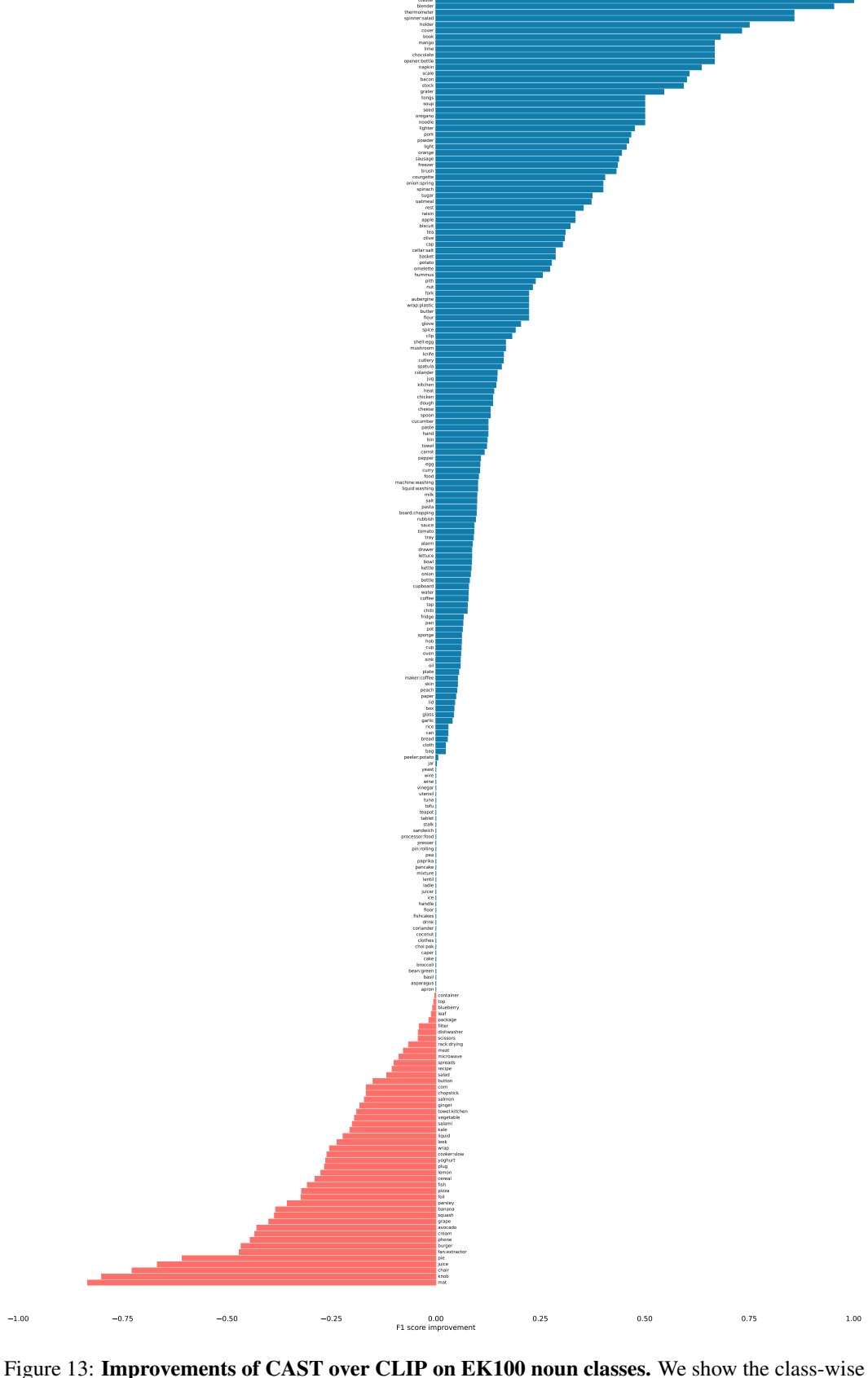

Figure 13: **Improvements of CAST over CLIP on EK100 noun classes.** We show the class-wise F1 score improvement of CAST over CLIP. CAST achieves an improvement of 7.9 points on average. Best viewed with zoom and color.

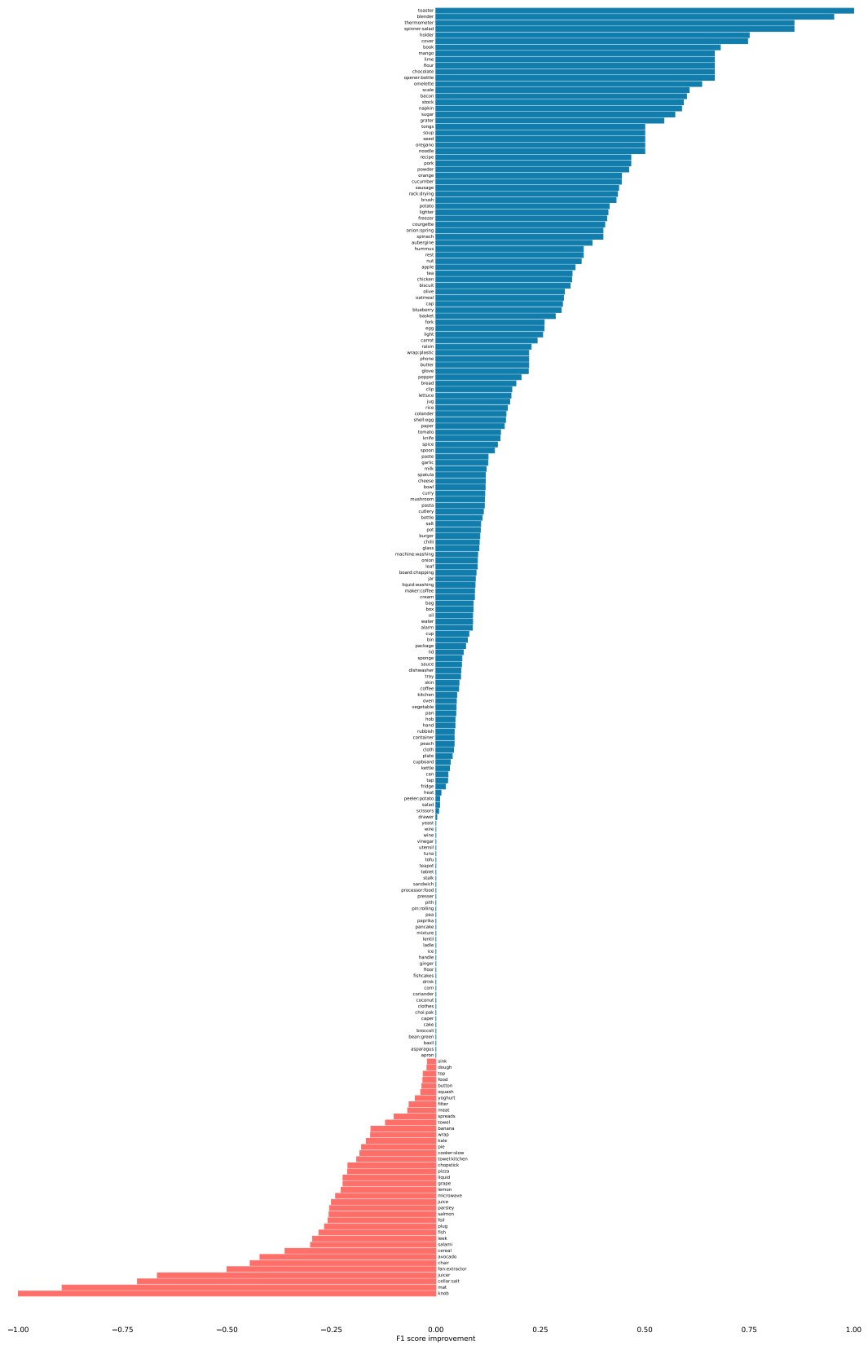

Figure 14: **Improvements of CAST over VideoMAE on EK100 noun classes.** We show the class-wise F1 score improvement of CAST over VideoMAE. CAST achieves an improvement of 9.2 points on average. Best viewed with zoom and color.

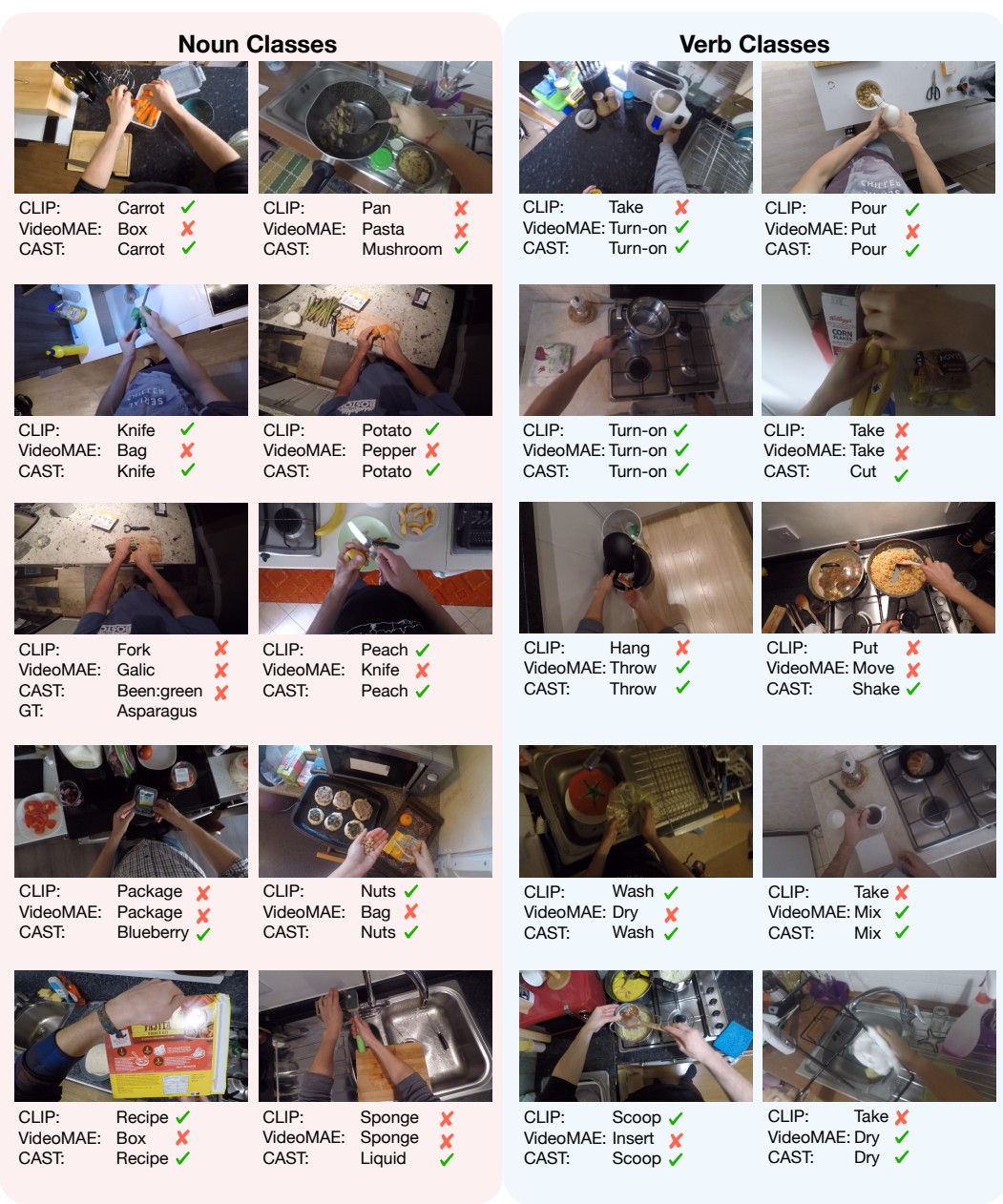

Figure 15: **Qualitative examples from EK100 comparing CLIP, VideoMAE, and the proposed CAST.** Each expert model shows more accurate predictions in their expertise but shows weaker performance on the other task. However, the proposed CAST consistently shows correct predictions for both noun and verb classes, demonstrating the effectiveness of the proposed spatio-temporal cross-attention mechanism. Best viewed with zoom and color.

