# OpenReview forum: "CAST: Cross-Attention in Space and Time for Video Action Recognition"
_NeurIPS.cc/2023/Conference — NeurIPS 2023 poster_

### Official Review · Reviewer_rpNG · 2023-07-05

**Soundness:** 4 excellent
**Presentation:** 4 excellent
**Contribution:** 4 excellent
**Rating:** 7
**Confidence:** 4

**Summary:**

This paper proposes a novel two-stream architecture called Cross-Attention in Space and Time (CAST) for video action recognition. The proposed architecture achieves a balanced spatio-temporal understanding of videos using only RGB input.

**Strengths:**

1. The paper is well-written and well-structured.
2. The implementation of Adapters in the cross-attention mechanism is impressive.
3. The paper presents sufficient experiments.

**Weaknesses:**

In Table 2, when compared to VideoVAE, CAST only achieve minor improvement on the SSv2 dataset. The authors are encouraged to discuss why the cross-attention mechanism appears to have a limited impact on enhancing performance for motion-dominated datasets, like SSv2.  Likewise, it would be valuable to have an analysis explaining why the proposed method underperforms compared to AIM (CLIP pretrained only) on the K400 dataset.

**Questions:**

See weakness

**Limitations:**

The authors have addressed the limitations

---

> ### Author Rebuttal · Authors · 2023-08-09
>
> We thank the reviewer for the great questions. We address the issues raised by the reviewer below.
>
> * Why does CAST show minor improvement on the SSV2 dataset compared to VideoMAE?
>
>     The main reason for the minor improvement is that the SSV2 is an object appearance agnostic action dataset. SSV2 consists of actions such as “pushing something from left to right”, “pushing something from right to left”, “throwing something”, “squeezing something”, and so on. Temporal experts could show strong performance in discriminating such movements, not specific objects. Given that VideoMAE already serves as a strong temporal expert, adding fine-grained spatial information from the spatial expert (CLIP) may not provide significant additional benefits for this specific dataset.
>
>     It is crucial to clarify that the primary objective of our work is not to achieve the best performance on a particular dataset. Instead, our focus is on achieving a balanced spatio-temporal understanding, which results in good performance across datasets with diverse characteristics. As evident in our results, CAST demonstrates strong performance in this respect, achieving a harmonic mean of 71.5% across EK100 verb, noun, SSV2, and K400 datasets. This demonstrates the effectiveness of CAST in achieving balanced performance across different action recognition tasks and datasets.
>
> * Why the proposed method underperforms compared to AIM on the K400 dataset?
>
>     We have investigated the reason for the lower performance of CAST compared to AIM on the K400 dataset. We found that the reason is mainly due to the preprocessing steps. In prior works such as VideoMAE, the videos are resized so that the shorter side is 320 pixels, and then random crops are taken from the resized video. However, in our previous implementation, we resized the videos so that the shorter side is 256 pixels. This discrepancy in preprocessing accounts for the lower performance of CAST compared to AIM on the K400 dataset. When we use the same video resizing as prior works, we achieve 85.4% accuracy on the K400 dataset, outperforming AIM-B (84.5%).

---

### Official Review · Reviewer_TFT9 · 2023-07-05

**Soundness:** 3 good
**Presentation:** 3 good
**Contribution:** 3 good
**Rating:** 6
**Confidence:** 3

**Summary:**

This paper presents an approach to action recognition that is based on the fusion of two streams of analysis. This involves a spatial stream and a temporal stream that interact through a novel mechanism to improve classification performance.

**Strengths:**

1. The paper is generally well written

2. The cross-attention mechanism is quite novel with respect to methods for 2 stream fusion.

3. Results appear to be quite good on standard benchmarks, and best when considering average performance

**Weaknesses:**

1. The B-CAST mechanism is quite complex and hard to unpack. With that said, I'm not sure there is a simpler way of framing this, but I would encourage the authors to consider simplifying the description (if possible). Figure 3 is fantastic in this regard.
2. The results only marginally improve the state of the art and are second best on the two main benchmarks considered, but bested by different models in each case. I wouldn't hold the paper back due to this, but it makes the contribution a bit weaker.

**Questions:**

1. Compared to other models, the proposed model appears to be better at verbs and weaker at nouns. Is it possible for the authors to comment on that trade-off?

2. Figure 6 shows a few good illustrated examples. Would the authors consider presenting more examples (e.g. as Supplementary Material)?

**Limitations:**

There is recognition of some of the limitations of the work throughout the paper, but not a dedicated statement.

---

> ### Author Rebuttal · Authors · 2023-08-09
>
> We thank the reviewer for the constructive suggestions. We address the issues raised by the reviewer below.
>
> * Straightforward unpacking of B-CAST mechanism
>
>     To facilitate a straightforward understanding of the B-CAST architecture, we present two additional figures in the global response PDF attached. Figure 1 illustrates how the model transforms and computes features in detail with tensor dimensions. After passing the previous layer output through the MHSA of each expert, we reduce the feature dimension by half with a down projection layer followed by layer normalization. Subsequently, the two experts exchange features, reshaping them according to the window shape. After the cross-attention, we upscale the feature by a factor of two using an up-projection layer. For an in-depth understanding of tensor dimensions, we direct the reviewer to Table 1 of the supplementary material.
>
>     Figure 2 demonstrates where the proposed cross-attention mechanism attends to. In Temporal-to-Spatial (T2S) cross-attention, the query corresponds to a spatial patch of the spatial expert. In T2S, the query attends to temporal patches at the same position across all frames. We term this cross-attention window shape as “time”. Conversely, in Spatial-to-Temporal (S2T) cross-attention, the query represents a patch of the temporal expert. In S2T, a temporal query attends to all spatial patches of the same frame. We term this cross-attention window shape as “space”. We empirically find that using T2S and S2T shows favorable performance compared to alternative choices in Table 1 (e) of the main paper.
>
> * Performance gap
>
>     We have investigated the reason for the relatively marginal performance improvement of CAST compared to state-of-the-art methods on the K400 dataset. We found that the reason is mainly due to differences in the preprocessing steps. In prior works such as VideoMAE, the videos are resized so that the shorter side is 320 pixels, and then random crops are taken from the resized video. However, in our previous implementation, we resized the videos so that the shorter side is 256 pixels. This discrepancy in preprocessing accounts for the lower performance of CAST compared to AIM on the K400 dataset. When we use the same video resizing as prior works, we achieve 85.4% accuracy on the K400 dataset and 77.9% harmonic mean of SSV2 and K400 accuracies surpassing the previous state-of-the-art (77.2%). On the EK100, the previous state-of-the-art method OMNIVORE trained with an external video dataset, K400, shows 49.9% action accuracy. When we train CAST with an external video dataset (Wang et al., 2022 [34] of supplementary material), we achieve 50.2% action accuracy as shown in Table 8 of supplementary material.
>
> * Verb-Noun performance trade-off on EK100
>
>     In comparison to OMNIVORE (with 69.5% accuracy for verbs and 61.7% for nouns) and MTV-HR (with 68.0% accuracy for verbs and 63.1% for nouns), CAST exhibits a larger gap between verb and noun prediction accuracies, achieving 72.7% accuracy for verbs and 60.6% for nouns. This can be attributed to the fact that the noun prediction task in the EK100 dataset entails a highly detailed 300-way classification of kitchen objects, representing a fine-grained classification challenge. Both MTV-HR and OMNIVORE are pre-trained on image datasets as well as video datasets (MTV-HR pre-trained on the ImageNet-21K and K400, and OMNIVORE pre-trained on the ImageNet-21K, ImageNet-1K, SUN-RGBD, and K400), which contributes to their superior performance in fine-grained noun classification. In contrast, CAST utilizes a spatial expert that is based on CLIP, an image-text contrastively pre-trained model. It is plausible that CLIP, despite its pre-training, may exhibit relative weaknesses in fine-grained noun classification compared to MTV-HR and OMNIVORE. Importantly, MTV-HR employs high-resolution image pre-training. Notably, similar to CAST, other CLIP-based methods like ST-Adapter and AIM also demonstrate relatively wider gaps between verb and noun accuracies. ST-Adapter yields 67.6% accuracy for verbs and 55.0% for nouns, while AIM achieves 64.8% accuracy for verbs and 55.5% for nouns. CAST surpasses these approaches through the effective bi-directional cross-attention-based information exchange between spatial and temporal experts, ultimately resulting in stronger performance across both verb and noun prediction tasks. We believe that this gap can be reduced by employing a spatial expert model with enhanced fine-grained classification capabilities, for instance, one pre-trained with higher-resolution images as seen in MTV-HR.
>
> * More qualitative examples
>
>     We show more qualitative examples in Fig. 9 of supplementary material.
>
> * Limitations
>
>     We have dedicated limitation and broader impacts sections in the supplementary material.

---

### Official Review · Reviewer_PRLf · 2023-07-06

**Soundness:** 1 poor
**Presentation:** 2 fair
**Contribution:** 2 fair
**Rating:** 6
**Confidence:** 5

**Summary:**

This paper presents two-stream vision transformers, dubbed CAST, for balanced spatiotemporal video representation learning. Given the two experts, CLIP [36] and VideoMAE [46] for spatial and temporal expert, the proposed B-CAST module allows the exchange of complementary information across the separte experts via MHCA. Such information exchange appears to be crucial for achieving balanced performances on appearance-centric (K400, Noun prediction in EK) and motion-centric benchmarks (SSv2, Verb prediction in EK).

**Strengths:**

1.	The paper deliberately engineers to integrate two separate research streams, adapter [34, 60] and representation fusion [12, 33, 59], into one framework for balance spatiotemporal video representation learning.
2.	The proposed method achieves strong performances on both appearance- and motion-centric benchmarks.
3.	Extensive investigation with several instantiations for CAST conducted in the ablation studies.


**Weaknesses:**

1.	In Tab. 1b, why are the three baseline models fully fine-tuned without adapters? Given that a partially tuned model with adapters is reported to be more effective than a fully fine-tuned model [34, 60], the comparison seems unfair. Reevaluating these models with adapters would make for a more fair comparison.
2.	Concerning the first point, are the model "independent experts" in Tab. 1a partially tuned with adapters or fully fine-tuned as in Tab. 1b? If it's the latter, for a fair comparison, CAST should be compared with an ensemble of two experts partially trained with adapters rather than with fully fine-tuned ones.
3.	For a more in-depth understanding, I suggest including the number of parameters, FLOPs, and, if possible, inference throughput in Tab. 1, particularly in Tabs. 1a, 1b, and 1d..
4.	The proposed method appears to increase computational cost and training & inference speed. Please provide a comparison of FLOPs, training & inference throughput with the baseline experts [36, 46], their ensemble (“independent experts” in Tab.1a), and other adapter-based methods [34, 60]. This would allow readers to comprehensively evaluate the applicability of the proposed method.
5.	In Tab.1e, there is only a small performance gap between between CAST (4th row) and the model in 3rd row, i.e, (T2S, S2T) = (space, time). Does this imply that, under conditions where attention is optimized (L278), the important factor is layer-wise mixing of the features from the two experts, rather than the specific method of mixing? For your reference, I consider the "Lateral" model in Tab.1b, which linearly fuses each other, to be an unfair comparison to the models in Tab.1e due to its less computation. It would be better to elaborate the discussion including results of (T2S, S2T) = (space, time), (time, time) or other variants.
6.	Table 2 should include information on the pretraining dataset, FLOPs, # trainable parameters, # frames, and, if possible, inference speed for a comprehensive comparison.


**Questions:**

1.	Is bi-directional information fusion necessary? What would be the result if we drop one of T2S MHCA or S2T MHCA?
2.	Has any analysis been conducted on verb prediction results similar to the approach taken in Fig. 5?
3.	(Minor) There is a discrepancy between the captions in Figure 4 and the actual subfigures. As a result, the text in lines 188-195 and 204-212 should be adjusted to refer to the correct subfigures.


**Limitations:**

The authors have provided a comprehensive discussion on the potential limitations and social impact of their work in Supp. 8 and 9. If the proposed model is found to require more computational cost or slows the processing speed down due to two-stream architecture, this inefficiency should also be considered as a limitation, which would pave the way for future improvements.

[Justification of the rating]
I acknowledge that this paper deliberately integrates two separate methods, adapter [34, 60] and feature fusion [12, 33, 59], into one module for balance spatiotemporal video representation learning. However, I’m concerned with the fairness of the experiment (refer to W1, W2, W3, and W6) and efficiency (W4). Consequently, my preliminary rating leans towards “borderline reject,” but I remain open to changing the score if the authors provide adequate discussions during the rebuttal period.

[Post-rebuttal justification]
Most of my concerns are well addressed by the rebuttal. I've gone through the rebuttal and found the provided experimental results seems to be reasonable and solid. I strongly recommend adding these results and the related discussions to the final manuscript. Consequently, I raise my rating to "weak accept."

---

> ### Author Rebuttal · Authors · 2023-08-09
>
> **Fair comparison**
>
> For a fair comparison with baselines, we augment Table 1 (a), (b), and (d) of the main paper with total parameters, GFLOPs/View, and throughput.
>
> ***Fusion baselines***
> For a fair comparison with baseline information exchange methods, we add adapters to add, concat, and lateral fusion baselines. We fine-tune the adapters only. We show the results in Table below.
>
> |Method|Late|Layer-wise|Total Param(M)|Tuneable Param(M)|GFLOPs/View|Throughput(V/s)|EK100 Verb|EK100 Noun|EK100 Act|
> |-|-|-|-|-|-|-|-|-|-|
> |Add|✅||187|15|343|34|68.9|56.6|44.2|
> |Concat|✅||188|16|343|34|69.2|56.4|44.5|
> |Lateral||✅|201|29|366|32|68.9|49.1|39.0|
> |CAST||✅|217|45|391|28|72.5|60.3|48.7|
>
> CAST shows significant improvement over all the baselines with adapters. CAST shows more than 4 points improvement compared to the baselines.
>
> ***Indep. experts baselines***
> For a comprehensive comparison, we present the results of independent experts with and without adapters in the table below.
> |Method|Total Param(M)|Tuneable Param(M)|GFLOPs/View|Throughput (V/s)|EK100 Verb|EK100 Noun|EK100 Act|
> |-|-|-|-|-|-|-|-|
> |Indep. experts w/o adapter|172|172|321|38|70.7|50.1|40.0|
> |Indep. experts w/ adapter|187|15|343|34|68.1|54.2|41.7|
> |Ensemble of experts w/ adapters|188|15|343|33|68.2|55.3|42.9|
> |CAST|217|45|391|28|72.5|60.3|48.7|
>
> CAST surpasses all the baselines compared. Most importantly, CAST achieves an impressive 5.8-point boost over the ensemble experts with adapters baseline.
>
> We observe that CAST shows a quite good trade-off between the performance and computation, compared to the baselines compared.
>
> ***Projection ratio***
> We choose the projection ratio of 1/2 as it yields the best trade-off between the computation and action accuracy.
>
> |Method|Total|Tuneable|GFLOPs/View|Throughput(V/s)|EK100 Verb|EK100 Noun|EK100 Act|
> |-|-|-|-|-|-|-|-|
> |1/8|191|19|351|31|70.7|59.9|47.4|
> |1/4|198|26|361|30|71.3|59.8|47.4|
> |1/2|217|45|391|28|72.5|60.3|48.7|
> |1|275|103|483|24|72.1|59.8|48.6|
>
>
> **Comprehensive comparison with existing methods**
>
> We compare the computational cost of CAST with independent expert models and other adapter-based models, presenting the results in the table below. Among these, CLIP-B/16 with adapter represents a partially tuned spatial expert, while VideoMAE-B/16 with adapter signifies the temporal expert.
>
> ***Training:***
> We report the single-step time (including forward pass, backward pass, and parameter update) using the EK100 dataset on a single GPU (RTX 3090) with 24GB of memory. To ensure a fair comparison, we exclude data loading and distributed communication from the step time measurement. We maintain an equal batch size of 6/GPU across all models.
>
> ***Inference:***
> We measure latency with a batch size of 1, focusing solely on the forward pass time. We also measure throughput with a batch size of 32.
>
> |Method| Total Param(M) | Tune Param(M) | Frames | Step Time (S)| VRAM (GB)|GFLOPs/View|Latency (S)|Throughput(V/s)|EK100 Verb|EK100 Noun|EK100 Act.
> |-|-|-|-|-|-|-|-|-|-|-|-|
> | CLIP-B w/ adapter |93|7|8|0.14|6.6|152|13.7|97|54.8|54.8|35.0|
> | VideoMAE-B w/ adapter|94|7|16|0.21|13.0|192|20.1|57|68.4|48.1|38.2|
> | ST-Adapter-B|93|7|16|0.31|12.6|-|22.9|49|67.6|55.0|-|
> | AIM-B|97|11|16|0.35|12.7|405|31.5|37|64.8|55.0|41.3|
> | CAST-B|217|45|16|0.44|21.0|391|48.2|28|72.5|60.3|48.7|
>
> **Attention window shape**
>
> We show the exhaustive search results in the table below.
>
> | Window shape T2S  | Window shape S2T | Tune param(M) |  Verb| Noun  | Act. |
> |-|-|-|-|-|-|
> | time | space-time | 57 | 71.5 | 60.9 | **48.7** |
> | time | space | 45 | 72.5 | 60.3 | **48.7** |
> | time | time | 43 | 72.1 | 60.6 | 48.6 |
> | space | space-time | 59 | 71.5 | 60.3 | 48.4 |
> | space | space | 47 | 72.3 | 60.2 | 48.5 |
> | space | time | 45 | 72.0 | 60.2 | 48.5 |
> | space-time | space-time | 72 | 71.0 | 59.3 | 47.2 |
> | space-time | space | 59 | 71.9 | 60.3 | 48.4 |
> | space-time | time | 57 | 71.3 | 60.3 | 48.2 |
>
> The experimental results show that the performance is not quite sensitive to the window shape. We choose the time attention for T2S and space attention for S2T, as this configuration offers a good trade-off between the number of learnable parameters and action accuracy.
>
>
> **Augmenting Table 2.**
>
> We will add the information on the pretraining dataset, FLOPs, the number of trainable parameters, the number of frames, and inference speed in Table 2 of the revised paper.
>
>
> **Bi-directional cross-attention**
>
> To validate the effectiveness of bi-directional cross-attention, we compare it with uni-directional cross-attention by individually omitting S2T and T2S in the table below.
>
> | Method | Total Param(M) | Tune Param(M) | Throughput (V/s) | Verb | Noun | Act. |
> |-|-|-|-|-|-|-|
> | Indep. experts w/ adapter | 187 | 15 | 34 | 68.1 | 54.2 | 41.7 |
> | S2T only | 210 | 38 | 30 | 71.2 | 55.0 | 43.7  |
> | T2S only | 208 | 36 | 30 | 68.7 | 60.5 | 46.7  |
> | CAST | 217 | 45 | 28 | 72.5 | 60.3 | 48.7  |
>
> CAST's bi-directional cross-attention outperforms uni-directional cross-attention. CAST achieves a 5-point enhancement over the S2T-only baseline and a 2-point improvement over the T2S-only baseline.
>
> **Category-wise analysis of verb prediction performance**
>
> We have conducted an analysis of the verb prediction performance across different verb categories, not super classes, and we present the results are in Figures 5 and 6 as well as in Section 6 of the supplementary material. There is no super class definition for verbs in the EK100 dataset.
>
>
> **Missing caption in Figure 4.**
>
> We will include the missing caption for Figure 4 (c) and make the necessary revisions to the text in lines 188-195 and 204-212 accordingly in the final version.
>
>
> **Limitations**
>
> In the final version, we will acknowledge the efficiency reduction as a potential limitation.

---

> > ### Comment · Reviewer_PRLf · 2023-08-16
> > **Response to the authors**
> >
> > I appreciate the authors for providing the rebuttal. I've gone through the rebuttal and found the provided experimental results appear to be reasonable and solid. I highly recommend revising the paper to incorporate these results and the related discussions from the rebuttal. Consequently, I raise my rating to "weak accept."

---

### Official Review · Reviewer_5TeX · 2023-07-06

**Soundness:** 3 good
**Presentation:** 3 good
**Contribution:** 2 fair
**Rating:** 6
**Confidence:** 5

**Summary:**

The manuscript proposes an approach of action recognition. The key idea behind the proposed approach is the usage of a two stream network where one network is specialized to encode the spatial details while the other to encode the temporal details. The approach also presents additional cross connections to effectively integrate the information from one stream to another. Experiments on three benchmark datasets shows that the proposed approach is capable of outperforming existing approaches on datasets requiring spatial reasoning, temporal reasoning and spatio-temporal reasoning.

**Strengths:**

The paper is written clearly explaining the various contributions and the underlying motivations. Majority of the existing approaches consider action recognition as a generic problem. However, the analysis presented in the manuscript showing the performance variations of existing approaches on datasets requiring spatial reasoning, temporal reasoning and spatio-temporal reasoning is quite intuitive and useful to the research community. The various design choices are validated with an extensive set of ablation studies. This includes the impact of the various backbones, pretraining strategies and the contributions. The improved performance on the different datasets compared to existing approaches showcases the effectiveness of the proposed approach. Moreover, this work could inspire the action recognition research community to move away from the paradigm of one model for all.

**Weaknesses:**

Two stream networks for action recognition is not a novel concept. For instance there are works that use RGB input in one network and optical flow in the other [1,2], two RGB input streams sampled at different frame rates [3] or spatial resolution [4], etc. However, there is no discussion regarding these approaches in the manuscript. The work of [5] is also similar to the proposed approach as it uses two different encoders for extracting spatial and temporal features. Even though the proposed approach is not exactly derivative of the few above-mentioned approaches, it is highly recommended to include a discussion comparing these approaches.

In order to predict the final action class, the adapter output of the CLS token of spatial expert and the adapter output of the global average pooled features of temporal expert is added followed by a classification layer. This addition operation across the features may diminish the discriminative information present in the individual streams. Instead, one may predict the action logits separately followed by an average fusion of the logits. Even though this operation is the same as the one presented in the paper, the compute graph is different and hence the gradient flow.

Even though the proposed approach results in improved performance compared to existing approaches, the comparison sometimes seems unfair. For example, the spatial expert of the best CAST model is initialized with CLIP embedding while the temporal expert is initialized with VideoMAE pretrained on SSV2. The same goes for other datasets as well. Could the performance improvement be due to this additional data? Agreeing the fact this is the main concept of the proposed approach, the compute requirements needed for this additional pretraining also needs to be taken into context. Even though the computational complexity in terms of FLOPs is reported in the supplementary material, run time latencies could tell a different story. The reviewer recommends the authors to compare the run time latencies of the proposed approach with existing approaches. The number of trainable parameters reported in the supplementary material is also not indicative of how heavy the model is. It is recommended to report the total number of parameters present in the final model. This will be helpful to researchers while selecting a model by taking into consideration the compute/size vs accuracy tradeoff.

[1] Simonyan, Karen, and Andrew Zisserman. "Two-stream convolutional networks for action recognition in videos." Advances in neural information processing systems 27 (2014).

[2] Feichtenhofer, Christoph, Axel Pinz, and Andrew Zisserman. "Convolutional two-stream network fusion for video action recognition." Proceedings of the IEEE conference on computer vision and pattern recognition. 2016.

[3] Feichtenhofer, Christoph, et al. "Slowfast networks for video recognition." Proceedings of the IEEE/CVF international conference on computer vision. 2019.

[4] Fan, Quanfu, et al. "More is less: Learning efficient video representations by big-little network and depthwise temporal aggregation." Advances in Neural Information Processing Systems 32 (2019).

[5] Jiang, Bo, et al. "Two-pathway transformer network for video action recognition." 2021 IEEE International Conference on Image Processing (ICIP). IEEE, 2021.

**Questions:**

Please see the weaknesses section for detailed queries.

What is the CLS(.) operation in line 126?


**Limitations:**

The manuscript adequately addressed the limitations.

---

> ### Author Rebuttal · Authors · 2023-08-09
>
> **Related work**
>
> CAST is similar to two-stream networks [1,2] as it learns two streams for spatial and temporal context. Unlike these approaches involving optical flow estimation, CAST uses only RGB streams. Similar to CAST, SlowFast networks [3] and bLVNet-TAM [4] also use only RGB streams. SlowFast employs slow and fast streams for spatial and temporal information but differs from CAST's transformer-based architecture.
>
> CAST and bLVNet-TAM [4] have distinctions. CAST aims for balanced spatio-temporal understanding, while bLVNet-TAM prioritizes computation efficiency, using different stream resolutions. CAST's unique design features bi-directional cross-attention, while bLVNet-TAM relies on temporal shifting.
>
> Jiang et al. [5] propose uni-directional cross-attention between RGB and edge encoders. CAST's bi-directional cross-attention in bottleneck architecture achieves effective fusion. This CAST architecture outperforms discussed methods [3-5], as shown in the tables below.
>
> | Method       | Backbone    | SSV2  | K400 |
> |-|-|-|-|
> | SlowFast     | ResNet-101  | -     | 79.8 |
> | bLVNet-TAM   | bLResNet-50 | 65.2  | 73.5 |
> | CAST         | CAST-B      | 71.6  | 85.4 |
>
> We will add the missing references [4,5] as suggested, building upon the reviewer's insightful feedback that enhances our analysis and comparisons with related works [1-3].
>
> **Classification head architecture**
>
> We have adopted the recommended head architecture, wherein we independently predict action logits for each expert and then average the logits, as depicted in Fig. 3 of the global response PDF. The results using this architecture are presented in the following table. For K400 experiments, we utilize a VideoMAE pre-trained on the hybrid dataset [34] as the temporal expert for both the methods compared.
>
> | Method     | SSV2          | K400        |
> |-|-|-|
> | add token  | 71.6          | 85.6        |
> | avg logit  | 71.8   | 85.4 |
>
> The suggested head architecture yields a minor 0.2-point improvement on the SSV2 dataset and a 0.2% point decline on K400 compared to the architecture used in the main paper. These results imply that discriminative information might already be learned through cross-attention in each layer.
>
> **Fair comparison: data-efficiency**
>
> Compared to AIM and ST-Adapter, CAST does not require extra training data. In our main setup, similar to AIM and ST-Adapter, we use the pre-trained CLIP model as the spatial expert. However, we utilize VideoMAE as the temporal expert, training it from scratch on the target dataset. For example, on EK100, we first self-train VideoMAE and then fine-tune CAST end-to-end. This means CAST only needs additional VideoMAE pre-training time on the target dataset, without any other extra requirements compared to AIM and ST-Adapter.
>
> In contrast to non-adapter-based methods like VideoMAE, CAST indeed demands more pre-training time and additional training data. For a fair comparison, we conducted an additional experiment, shown in Table below.
>
> | Method | Spatial Expert | Temporal Expert | EK100 Verb | EK100 Noun | EK100 Act. |
> |-|-|-|-|-|-|
> | VideoMAE | - | VideoMAE w/ EK100 | 70.5 | 51.4 | 41.7 |
> | AIM-B | CLIP w/ WIT400M    | - | 64.8 | 55.5 | 41.3 |
> | ST-Adapter-B | CLIP w/ WIT400M    | - | 67.6 | 55.0 | - |
> | CAST-B | ViT w/ IN-1K | VideoMAE w/ EK100   | 70.9 | 56.8  | 45.5 |
> | CAST-B | CLIP w/ WIT400M | VideoMAE w/ EK100   | 72.5 | 60.3  | 48.7 |
>
> In this experiment, we used a CAST variant with a spatial expert pre-trained on ImageNet-1K and a temporal expert pre-trained and fine-tuned on EK100. This variant achieved 45.5% action accuracy on EK100, surpassing AIM (41.3%) and VideoMAE (41.7%). Furthermore, this CAST variant also outperformed ST-Adapter, with accuracy values of 70.9% versus 67.6% for verb predictions and 56.8% versus 55.0% for noun predictions. Importantly, this variant required less pre-training data compared to AIM and ST-Adapter, making it more data-efficient. Although pre-training a ViT on ImageNet-1K took slightly more data and time than VideoMAE (EK100 consists of 11.5M frames while ImageNet-1K consists of 1M frames only, which corresponds to less than 10% of the EK100 dataset), the overall performance gain by the CAST variant justified this additional effort. These results highlight CAST's versatility and effectiveness, leveraging diverse pre-trained weights for competitive performance.
>
> **Fair comparison: computation efficiency**
>
> For a fair comparison with existing methods, we show the number of total parameters, inference-time throughput, and latency in the Table below.
>
> |Method|Spatial|Temporal|Total Parameters(M)|Throughput(V/s)|Latency(ms)|SSV2|SSV2 GFLOPs|K400|K400 GFLOPs|EK100 Verb|EK100 Noun|EK100 Act.|EK100 GFLOPs|
> |-|-|-|-|-|-|-|-|-|-|-|-|-|-|
> |AIM-B|CLIP w/ WIT400M|-|97|37|31.5|68.1|1238|84.5|1214|64.8|55.5|41.3|2430|
> |AIM-L|CLIP w/ WIT400M|-|354|9|122.5|69.4|5754|87.3|5604|-|-|-|-|
> |ST-Adapter-B|CLIP w/ WIT400M|-|93|49|22.9|69.3|977|82.5|911|67.6|55.0|-|-|
> |ST-Adatper-L|CLIP w/ WIT400M|-|-|-|-|71.9|4124|86.9|4124|-|-|-|-|
> |CAST-B|CLIP w/ WIT400M|VideoMAE w/ target data|217|28|48.2|71.6|2346|85.4|5865|72.5|60.3|48.7|2346|
>
> Compared to AIM and ST-Adapter, which employ a single expert model, our proposed CAST does not require additional pre-training data. The efficiency of the temporal expert, VideoMAE, is a key factor. VideoMAE's data-efficient nature is well-documented in its paper. Consequently, when compared to AIM and ST-Adapter, which rely on a sole expert, CAST surpasses AIM-L by +2.3% in performance. Additionally, CAST uses 137 million fewer parameters and computes ~3400 GFLOPs less than AIM-L on the SSV2. Similarly, compared to S-T Adapter-L, CAST achieves comparable performance with only a 0.3 point difference while computes ~1800 GFLOPs less on the SSV2.
>
> **CLS operation**
>
> The CLS operation refers to the process of extracting the CLS token.

---

> > ### Comment · Reviewer_5TeX · 2023-08-17
> > **Response to rebuttal**
> >
> > Most of the concerns raised in the initial review stage are addressed in the rebuttal. Hence I keep my original reccommendation of weak accept.

---

### Official Review · Reviewer_sJb4 · 2023-07-09

**Soundness:** 3 good
**Presentation:** 3 good
**Contribution:** 3 good
**Rating:** 6
**Confidence:** 4

**Summary:**

This paper proposes a method, namely CAST, for video action recognition based on adapting from large-scale pre-trained models (e.g., CLIP and VideoMAE). The main motivation is to balance and exchange information between spatial and temporal information of two different experts: spatial and temporal. The proposed CAST is an adapter architecture, which adopts CLIP as a spatial expert and VideoMAE as a temporal expert and introduces the cross attention to encourage interaction between these two streams. Experiments are conducted on Epic-KITCHEN-100, Kinetics-400, and Something-Something-v2 with good results compared with current methods. Written presentation is good and mostly easy to follow.

**Strengths:**

* The motivation of balancing between spatial and temporal information of video action recognition makes sense and the proposed architecture of CAST technically sounds.

* Experimental results are strong: (i) good results compared with existing methods; (ii) various ablation to justify the model design choices.

**Weaknesses:**

* Although the good improvements are shown, the downside may be: there may be an unfair comparison with some other adapter-methods, e.g., AIM where they built up on only one expert model. This means CAST needs more data / pretraining time (of CLIP and VideoMEA);

* There is no FLOPs (or runtime) comparison in table 2. As CAST employs a 2-stream architecture, there will be more computation compared with previous methods.

**Questions:**

* Some minor comments / typos:
- line 104: "p pixels" -> p x p pixels for more consistent with that in line 107?
- line 106: "BT x N x D" -> should be "B x TN x D" for more consistent with that in line 108?
- Figure 4 caption has no explanation for (c).
- line 204: Figure 4 (c) instead?

**Limitations:**

The reviewer does not foresee any potential negative societal impact of this work.

---

> ### Author Rebuttal · Authors · 2023-08-09
>
> We thank the reviewer for the great questions. We address the issues raised by the reviewer below.
>
> **Unfair comparison with adapter-methods**
>
> For a fair comparison, we conducted an additional experiment, as shown in Table 4 of the supplementary material. Below, we have included the table for your convenience.
>
> | Method         | Spatial Expert     | Temporal Expert     | EK100 Verb | EK100 Noun | EK100 Action |GFLOPs/View|
> |---------------|------------------|-------------------|----------|----------|----------|-|
> | AIM-B          | CLIP w/ WIT400M    | -                   | 64.8       | 55.5       | 41.3       |405|
> | ST-Adapter-B   | CLIP w/ WIT400M    | -                   | 67.6       | 55.0       | -          |-|
> | CAST-B         | ViT w/ IN-1K       | VideoMAE w/ EK100   | 70.9       | 56.8       | 45.5       |391|
> | CAST-B         | CLIP w/ WIT400M    | VideoMAE w/ EK100   | 72.5       | 60.3       | 48.7       |391|
>
> In this experiment, we use a variant of CAST with a spatial expert pre-trained on the ImageNet-1K and a temporal expert pre-trained and fine-tuned on the EK100. This variant achieved an action accuracy of 45.5% on EK100, surpassing both AIM (41.3%) and VideoMAE (41.7%). Furthermore, this CAST variant also outperforms ST-Adapter, with accuracy values of 70.9% versus 67.6% for verb predictions and 56.8% versus 55.0% for noun predictions. Importantly, this variant requires less pre-training data compared to AIM and ST-Adapter, making it more efficient in terms of data requirements. It is worth noting that while pre-training a ViT on ImageNet-1K requires slightly more data (EK100 consists of 11.5M frames while ImageNet-1K consists of 1M frames only, which corresponds to less than 10% of the EK100 dataset) and time compared to VideoMAE, the overall performance gain achieved by the CAST variant justifies this additional effort. These results demonstrate the versatility and effectiveness of CAST, as it can leverage pre-trained weights from different sources and still achieve competitive performance.
>
> **FLOPs comparison**
>
> We show FLOPs, the number of frames per clip, the number of test views, and learnable parameters in Tables 8, 9, and 10 of the supplementary material. We found the error in GFLOPs of CAST in the supplementary material and show the fixed GFLOPs in the following Table. The GFLOPs of the other methods compared in the supplementary material are correct.
>
> |Dataset|View|GFLOPs/View|GFLOPs|
> |-|-|-|-|
> |EK100|2x3|391|2346|
> |SSV2|2x3|391|2346|
> |SSV2|5x3|391|5865|
>
> CAST shows favorable spatio-temporal balanced understanding performance on the EK100, SSV2, and K400 even with lightweight configurations.
>
> **Tensor shape in lines 106 and 108**
>
> The notation for line 106 and line 108 is different: line 106 is "BT x N x D" because it is an image passing through a spatial path, while line 108 is "B x TN x D" because it is a video passing through a temporal path. The difference occurs because the spatial path treats each frame as an independent image. We show the process of cross-attention of different dimensions of tensors in Supplementary Table 1.
>
> **Typo and missing caption**
>
> Thank you for your suggestions. We will fix the typo and add the missing caption in the final version.

---

> > ### Comment · Reviewer_sJb4 · 2023-08-15
> > **Thank you for the rebuttal**
> >
> > The rebuttal addressed most of my concerns, I have upgraded my rating to "weak accept". Thank the author(s) for the additional experiments & clarification.

---

### Author Rebuttal · Authors · 2023-08-09

We express our gratitude to the reviewers for their valuable and constructive comments. We have taken careful consideration of the points raised by each reviewer. We address these concerns comprehensively. Additionally, we have included a supplementary PDF containing figures that further support our responses and explanations for each reviewer's comments.

---

### Decision · Program_Chairs · 2023-09-21

**Decision:**

Accept (poster)

**Comment:**

The paper received unanimous accept reviews, including 4x Weak accept and 1x accept. The core idea is to leverage two expert models, CLIP for spatial understanding, and VideoMAE for temporal understanding, along with an adapter architecture that introduces the cross attention between these two streams. The resulting model obtains strong performance on 3 standard datasets. All reviewers appreciate the clearly written paper, with solid experimentation and good results. ACs concur with reviewer consensus.